ARTICLES
# Systematic characterization of gene function in the photosynthetic alga *Chlamydomonas reinhardtii*

Friedrich Fauser[1,2,15], Josep Vilarrasa-Blasi[2,3,15], Masayuki Onishi[4,5], Silvia Ramundo[6], Weronika Patena[1,2], Matthew Millican[2], Jacqueline Osaki[2], Charlotte Philp[2], Matthew Nemeth[2], Patrice A. Salomé[7], Xiaobo Li[1,2,14], Setsuko Wakao[8,9], Rick G. Kim[2], Yuval Kaye[2], Arthur R. Grossman[2], Krishna K. Niyogi[8,9,10], Sabeeha S. Merchant[7,11], Sean R. Cutler[12], Peter Walter[6], José R. Dinneny[2,3 ✉], Martin C. Jonikas[1,2 ✉] and Robert E. Jinkerson[2,12,13 ✉]

**Most genes in photosynthetic organisms remain functionally uncharacterized. Here, using a barcoded mutant library of the model eukaryotic alga *Chlamydomonas reinhardtii*, we determined the phenotypes of more than 58,000 mutants under more than 121 different environmental growth conditions and chemical treatments. A total of 59% of genes are represented by at least one mutant that showed a phenotype, providing clues to the functions of thousands of genes. Mutant phenotypic profiles place uncharacterized genes into functional pathways such as DNA repair, photosynthesis, the CO$_2$-concentrating mechanism and ciliogenesis. We illustrate the value of this resource by validating phenotypes and gene functions, including three new components of an actin cytoskeleton defense pathway. The data also inform phenotype discovery in land plants; mutants in *Arabidopsis thaliana* genes exhibit phenotypes similar to those we observed in their *Chlamydomonas* homologs. We anticipate that this resource will guide the functional characterization of genes across the tree of life.**

Major contributions to our understanding of gene functions in photosynthetic organisms have been made by studying microbial models, including the discovery and characterization of the Calvin–Benson–Bassham CO$_2$ fixation cycle[1] as well as the structures[2], order[3] and cloning[4] of complexes in the photosynthetic electron transport chain. Advances in technology now provide opportunities for microbes to serve as powerful complements to land plants in the characterization of gene functions by enabling substantially higher experimental throughput[5].

The single-celled green alga *Chlamydomonas* (*Chlamydomonas reinhardtii*) is a well-established model system for studies of key pathways, including photosynthesis, primary metabolism, interorganelle communication and stress response[6]. Furthermore, amenability to microscopy and biochemical purifications have made *Chlamydomonas* a leading model system for studies of cilia[7–9]. Despite promising progress with the development of clustered regularly interspaced short palindromic repeats (CRISPR)-based reagents to generate targeted mutants[10,11], low editing efficiencies currently prevent large-scale CRISPR single guide RNA library screens in *Chlamydomonas*. The recent generation of a barcoded *Chlamydomonas* mutant collection facilitates the study of individual genes and enables forward genetic screens[12]. In the present work,

we leverage the amenability of *Chlamydomonas* to high-throughput methods to connect genotypes to phenotypes on a massive scale, allowing placement of genes into pathways and discovery of conserved gene functions in land plants.

## Results

**Systematic genome-scale phenotyping.** To connect genotypes to phenotypes, we measured the growth of 58,101 *Chlamydomonas* mutants representing 14,695 genes (83% of all genes encoded in the *Chlamydomonas* genome, based on the current genome annotation, v5.6) under 121 environmental and chemical stress conditions (both control and experimental conditions are given in Supplementary Tables 1 and 2). We pooled the entire *Chlamydomonas* mutant collection from plates into a liquid culture and used molecular barcodes to quantify the relative abundance of each mutant after competitive growth (Fig. 1a–f). Growth conditions included heterotrophic, mixotrophic and photoautotrophic growth under different photon flux densities and CO$_2$ concentrations, as well as abiotic stress conditions such as various pH and temperatures. We also subjected the library to known chemical stressors, including DNA-damaging agents, reactive oxygen species, antimicrobial drugs such as paromomycin and spectinomycin and the actin-depolymerizing drug

[1]Department of Molecular Biology, Princeton University, Princeton, NJ, USA. [2]Department of Plant Biology, Carnegie Institution for Science, Stanford, CA, USA. [3]Department of Biology, Stanford University, Stanford, CA, USA. [4]Department of Biology, Duke University, Durham, NC, USA. [5]Department of Genetics, Stanford University School of Medicine, Stanford, CA, USA. [6]Department of Biochemistry and Biophysics, University of California, San Francisco, San Francisco, CA, USA. [7]Department of Chemistry and Biochemistry and Institute for Genomics and Proteomics, University of California, Los Angeles, Los Angeles, CA, USA. [8]Department of Plant and Microbial Biology, University of California, Berkeley, Berkeley, CA, USA. [9]Molecular Biophysics and Integrated Bioimaging Division, Lawrence Berkeley National Laboratory, Berkeley, CA, USA. [10]Howard Hughes Medical Institute, University of California, Berkeley, Berkeley, CA, USA. [11]Department of Molecular and Cell Biology, University of California, Berkeley, Berkeley, CA, USA. [12]Department of Botany and Plant Sciences, University of California, Riverside, Riverside, CA, USA. [13]Department of Chemical and Environmental Engineering, University of California, Riverside, Riverside, CA, USA. [14]Present address: School of Life Sciences, Westlake University, Hangzhou, Zhejiang Province, China. [15]These authors contributed equally: Friedrich Fauser, Josep Vilarrasa-Blasi. ✉e-mail: dinneny@stanford.edu; mjonikas@princeton.edu; robert.jinkerson@ucr.edu

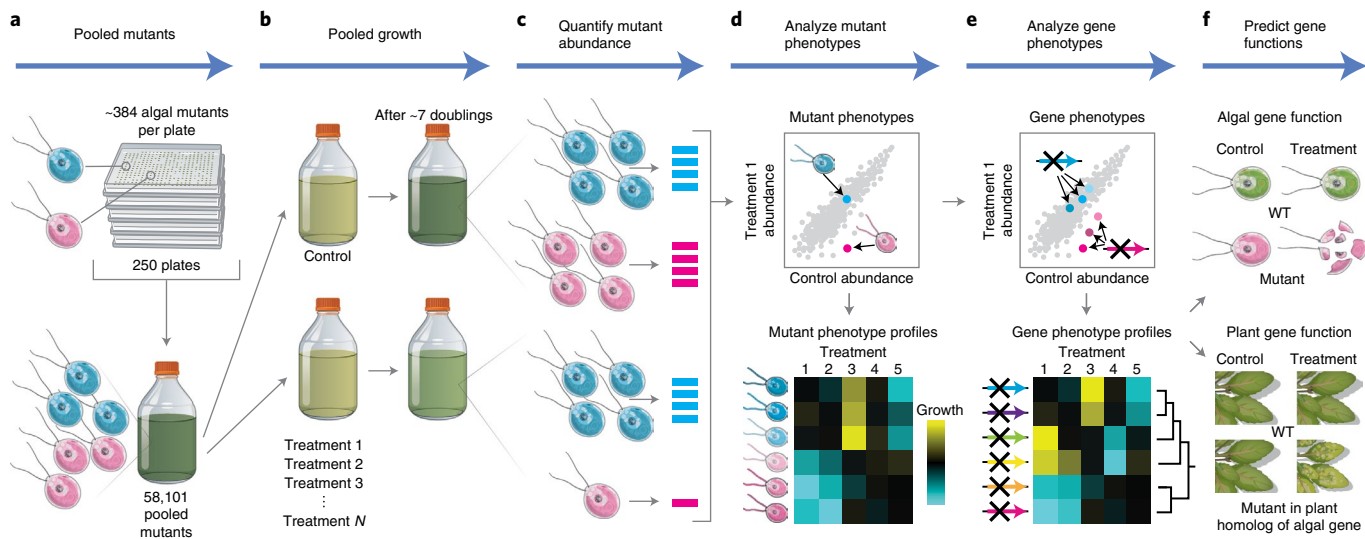

**Fig. 1 | Pooled phenotyping enables the large-scale discovery of genotype-phenotype relationships in a unicellular photosynthetic eukaryote. a**, The *Chlamydomonas* mutant library was pooled and used to prepare a liquid starting culture of 58,101 mutants. **b**, Aliquots of the starting culture were used to inoculate pooled growth experiments to assess the fitness of each mutant under a variety of environmental and chemical stress treatments. **c**, The relative abundance of each mutant was quantified via polymerase chain reaction (PCR)-based amplification of individual mutant barcodes and subsequent Illumina sequencing. **d**, Mutants negatively affected by the treatment have a lower barcode read count compared to the control. **e**, Many genes were represented by multiple mutants, which allowed the identification of high-confidence gene phenotypes. We then clustered genes based on their phenotypic profile to place genes into pathways and predict the functions of previously uncharacterized genes. **f**, The data predict gene function in *Chlamydomonas* and land plants. WT, wild type.

latrunculin B (LatB). To further expand the range of stressors in the dataset, we identified 1,222 small molecules from the Library of AcTive Compounds on Arabidopsis (LATCA)[13] that negatively influence *Chlamydomonas* growth (Extended Data Fig. 1, Supplementary Table 3 and Supplementary Data 1) and performed competitive growth experiments in the presence of 52 of the most potent compounds. We chose to screen the LATCA library for active compounds in *Chlamydomonas* because we believed that these compounds would be more likely to impact pathways both in *Chlamydomonas* and in plants, thus providing more general insights into gene functions in the green lineage. Taken together, this effort represents, to the best of our knowledge, the largest mutant-by-phenotype dataset to date for any photosynthetic organism, with 16.8 million data points (Supplementary Table 4).

**Mutants show genotype–phenotype specificity.** To identify mutants with growth defects or enhancements due to a specific treatment, we compared the abundance of each mutant after growth under the treatment condition to its abundance after growth under a control condition (Fig. 2a). We called this comparison a screen and the ratio of these abundances the mutant phenotype (Fig. 2b,c). Mutant phenotypes were reproducible between independent replicates of a screen (Fig. 2c,d).

Individual mutants exhibited genotype–phenotype specificity. For example, mutants disrupted in the DNA repair gene *POLYMERASE ZETA* (*POLZ*, encoded by Cre09.g387400) exhibited growth defects in the presence of the DNA crosslinker cisplatin, and these mutants did not show growth defects in unrelated screens (Fig. 2d). We observed similar genotype–phenotype specificity for other genes and phenotypes, including sensitivity to low $CO_2$, ciliogenesis and LatB sensitivity (Fig. 2d).

In many screens, mutants that exhibited phenotypes were enriched for disruptions in genes with expected function. In 46 out of 223 screens, at least one Gene Ontology (GO)[14] term was enriched (FDR < 0.05) in the genes disrupted in mutants whose growth was perturbed in the screen (Fig. 2e, Extended Data Fig. 2 and Supplementary Table 5). These enriched GO terms corresponded to functions known to be required for survival under the respective treatments. For example, screens with DNA-damaging agents resulted in GO term enrichments such as 'DNA replication', 'Nucleotide binding' or 'Damaged DNA binding'. These GO term enrichments suggest that the phenotypic screens are correctly identifying genes required for growth under the corresponding treatments.

In total, 10,380 genes (59% of all *Chlamydomonas* genes) are represented by one or more 5′ untranslated region (UTR), coding DNA sequence (CDS) or intron insertion mutant that showed a phenotype (decreased abundance below our detection limit) in at least one screen (Fig. 2f). Although a lone mutant showing a phenotype is not sufficient evidence to conclusively establish a gene–phenotype relationship, we anticipate that these data will be useful to the research community in at least three ways. First, they can help prioritize the characterization of candidate genes identified by other means, such as transcriptomics or protein–protein interactions. Second, they facilitate the generation of hypotheses about the functions of poorly characterized genes. Third, they enable prioritization of available mutant alleles for further studies, including to establish a gene–phenotype relationship by complementation and/or backcrossing. The genotype–phenotype specificity of individual mutants and the enrichment of expected functions suggest that our data can serve as a guide for understanding the functions of thousands of poorly characterized genes.

**High-confidence gene–phenotype relationships.** The availability of multiple independent mutant alleles for individual genes allowed us to identify high-confidence gene–phenotype relationships. When multiple independent mutant alleles for the same gene show the same phenotype, the confidence in a gene lesion–phenotype relationship increases, because it is less likely that the phenotype is due to a mutation elsewhere in the genome (on average, there are 1.2

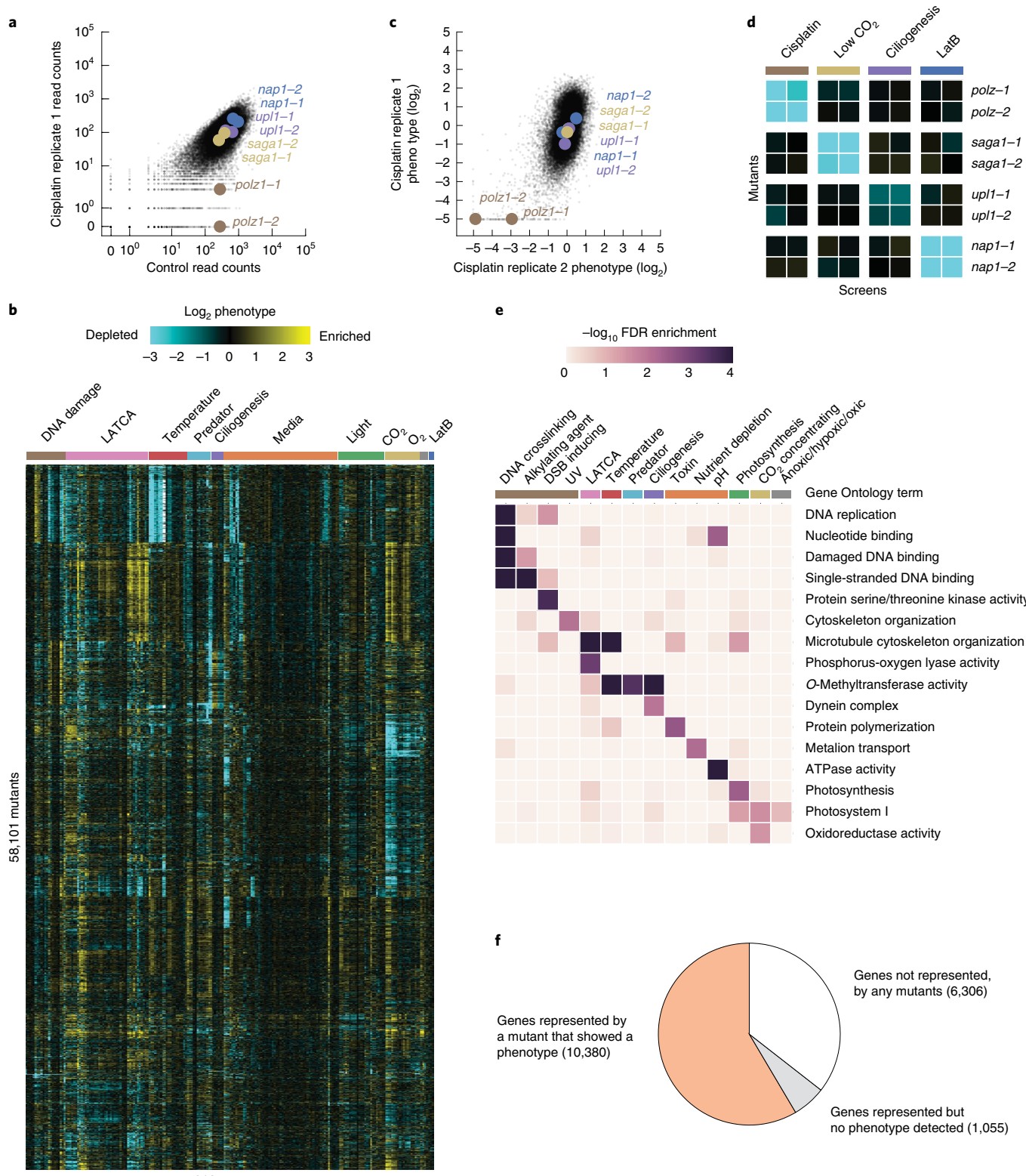

**Fig. 2 | We determined the fitness of 58,101 *Chlamydomonas* mutants under 121 growth conditions. a**, The phenotype of each mutant was determined by comparing its molecular barcode read count under a treatment and control condition. As an example, results from a screen using the drug cisplatin are shown. **b**, A hierarchically clustered heatmap shows the phenotype [$\log_2$(treatment reads/control reads)] of mutants across 212 screens representing 121 growth conditions. **c**, The typical reproducibility is illustrated with two replicate cisplatin screens. **d**, Mutants show screen-specific phenotypes. **e**, GO term analysis reveals enrichment of biological functions associated with specific screens. **f**, Most genes are represented by at least one mutant that shows a phenotype in at least one treatment condition. DSB, double-strand break; FDR, false discovery rate.

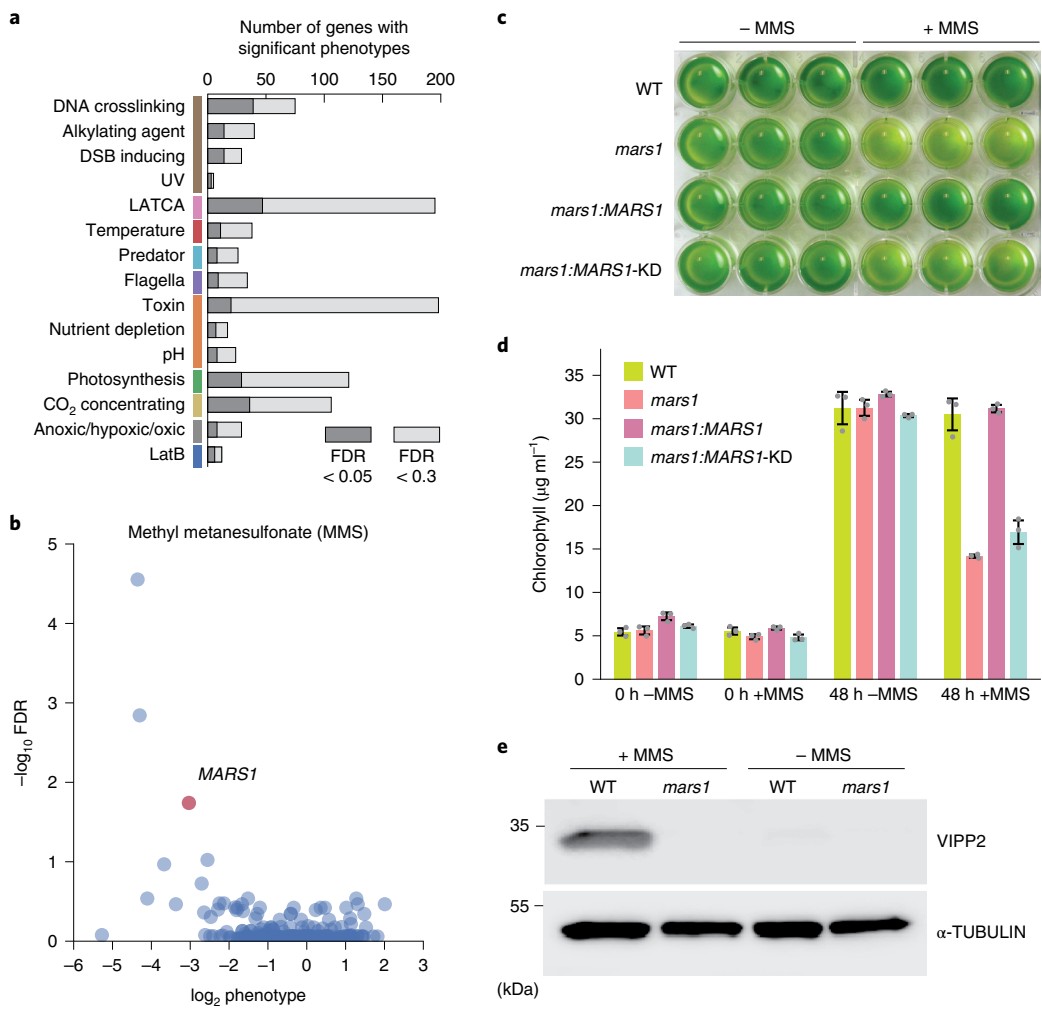

**Fig. 3 | Multiple alleles provide high confidence and reveal new phenotypes. a**, The number of genes with significant phenotypes in each class of screen is shown for two FDR thresholds. **b**, FDR is plotted against log$_2$ median phenotype for all genes in the MMS screen. **c**, Growth assay of WT, *mars1*, *mars1:MARS1* and *mars1:MARS1-KD* cells after 48 h in the presence or absence of MMS. Three biological replicates were used for each strain. For more details, Methods. **d**, Average chlorophyll concentrations of the liquid cultures shown in Fig. 3c. Data are presented as mean values ± s.d. $n = 3$ independent experiments. **e**, Immunoblot analysis of VESICLE-INDUCING PROTEIN IN PLASTIDS (VIPP2), a downstream target of MARS1, in WT and *mars1* cells in the presence or absence of MMS. Immunoblot representative of $n = 2$ independent experiments.

integration events per mutant, and the mutants can also carry other mutations such as point mutations) or that there was an error in mapping of the mutation[12]. Using a statistical framework that leverages multiple independent mutations in the same gene (Methods), we identified 1,218 high-confidence (FDR < 0.3) gene–phenotype relationships involving 684 genes (Fig. 3a and Supplementary Tables 6 and 7), including hundreds of genes with no functional annotation in the green lineage (Supplementary Table 8). Our gene–phenotype relationships include 302 high-confidence (FDR < 0.3) interactions involving 195 genes and 39 LATCA drugs, providing clues to the drugs' targets and improving the value of these drugs as tools for perturbing specific pathways (Supplementary Table 7). Based on the highest-confidence (FDR < 0.05) phenotypes, we suggest names for 89 previously unnamed genes (Supplementary Table 9).

As an example of how individual gene–phenotype relationships advance our understanding, we made the unexpected observation that mutants in the gene encoding the chloroplast unfolded protein response (cpUPR) kinase, MUTANT AFFECTED IN CHLOROPLAST-TO-NUCLEUS RETROGRADE SIGNALING (MARS1)[15], were sensitive (FDR < 10⁻⁹) to the DNA-damaging

agent methyl methanesulfonate (MMS) (Fig. 3b). We validated this phenotype in a separate growth assay and showed that the MMS sensitivity of these mutants is rescued by complementation with a wild-type copy of *MARS1*, but not by a kinase-dead version (Fig. 3c,d). We also determined that treatment with MMS led to induction of VESICLE-INDUCING PROTEIN IN PLASTIDS 2 (VIPP2), a highly selective cpUPR marker, in wild-type cells but not in mutants lacking *MARS1* (Fig. 3e). These results illustrate the value of our high-throughput data and suggest the intriguing possibility that the cpUPR is activated via MARS1 upon DNA damage or protein alkylation and has a protective role against these stressors.

**From phenotypes to pathways.** To facilitate data visualization and predict the functions of poorly characterized genes in our dataset, we used the principle that mutant alleles with similar phenotypes tend to occur in genes that function in the same pathway[5]. We clustered the 684 genes with high-confidence phenotypes based on the similarity of their phenotypes across different treatments (Fig. 4a and Supplementary Data 2). The correlation of phenotypes was

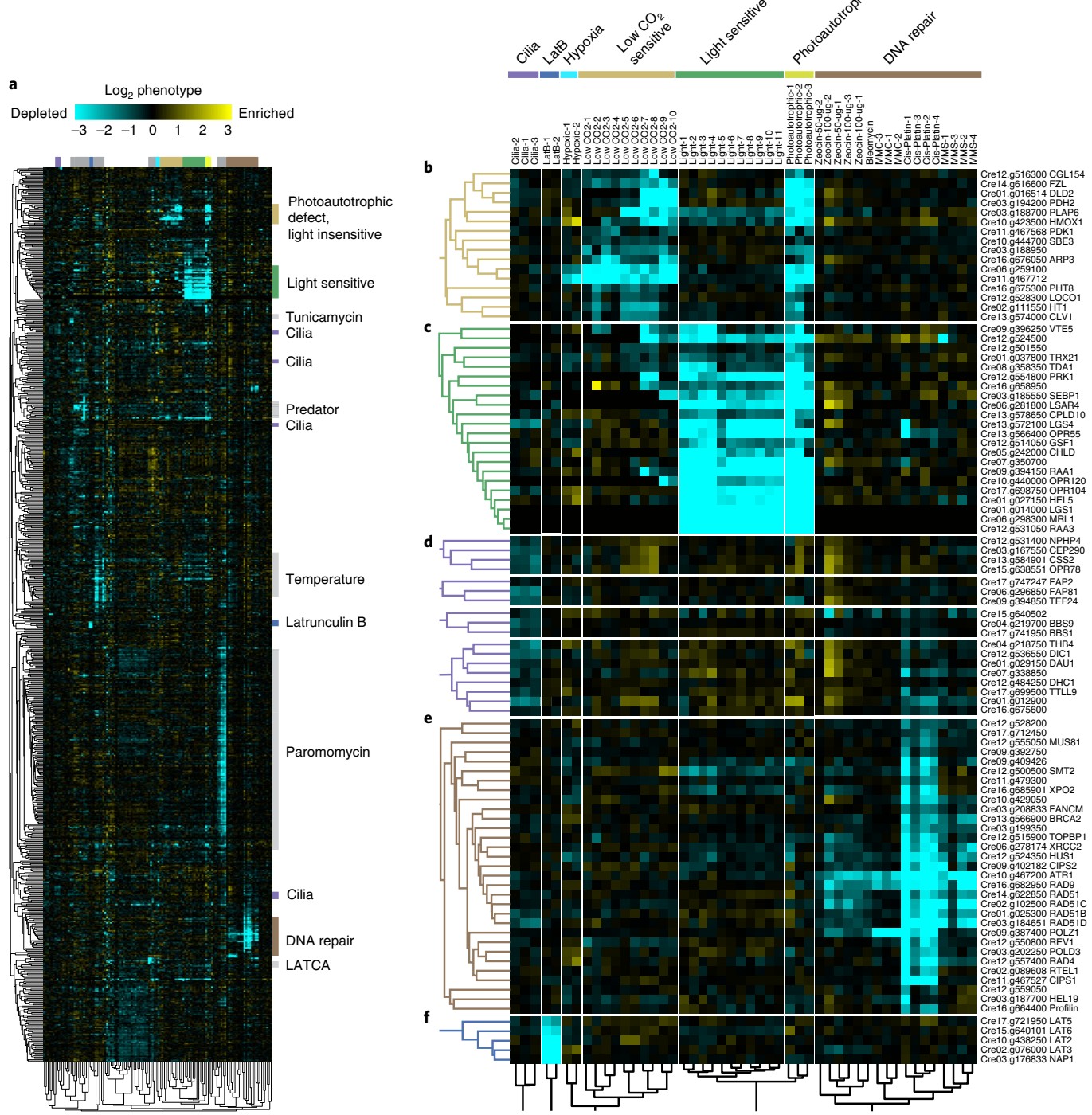

**Fig. 4 | Similarity of mutant phenotypes places genes into pathways and reveals new players. a**, A total of 684 genes were clustered based on the similarity of their phenotypes across 120 screens. **b–f**, Examples of how subclusters enriched in specific pathways predict new genes in these pathways: nonphotoautotrophic light insensitive (**b**), nonphotoautotrophic light sensitive (**c**), cilia (**d**), DNA damage sensitive (**e**) and LatB sensitive (**f**).

largely unrelated to transcriptional expression correlation[16], suggesting that the two approaches provide complementary information (Extended Data Fig. 3 and Supplementary Table 10). We named some of our gene clusters based on the presence of previously characterized genes or based on the conditions that produced the most dramatic phenotypes in a cluster (Fig. 4b–f and Supplementary Table 9). Below, we provide examples of how the data recapitulate known genetic relationships and provide insights into the functions of poorly characterized genes.

**Essential DNA repair pathways are conserved in green algae.** DNA damage repair pathways are among the best-characterized and most highly conserved across all organisms[17,18]; thus, they serve as a useful test case of the quality of our data. In our dataset, homologues of known DNA repair proteins are present in a large cluster (Fig. 4e), demonstrating the quality of our phenotypic data, validating our ability to identify that these genes work in a common pathway and extending the conservation of their functions to green algae.

Mutants for various DNA repair genes exhibit expected differences in their sensitivities to different types of DNA damage: (1) DNA double-strand breaks (zeocin and bleomycin), (2) DNA crosslinks (mitomycin C and cisplatin) and (3) DNA alkylation (MMS). For example, mutants exhibiting sensitivity to all DNA damage conditions included mutants lacking upstream DNA damage-sensing kinase ATAXIA TELANGIECTASIA AND RAD$_3$-related protein (ATR, encoded by Cre10.g467200) (ref. [19]), as well as mutants lacking the cell cycle checkpoint control protein RADIATION SENSITIVE 9 (RAD9, encoded by Cre16.g682950) or its binding partner HYDROXYUREA-SENSITIVE 1 (HUS1, encoded by Cre12.g524350) (ref. [20]). Mutants specifically sensitive to the double-strand break-inducing agents zeocin and bleomycin included the upstream sensor of double-strand breaks, the kinase ATAXIA-TELANGIECTASIA MUTATED (ATM, encoded by Cre13.g564350) (ref. [21]) (Supplementary Table 6) and DNA POLYMERASE THETA (POLQ, encoded by Cre16.g664301), which facilitates error-prone double-strand break repair and can maintain genome integrity when other repair pathways are insufficient[22,23] (Supplementary Table 6). Mutants specifically sensitive to the DNA crosslinker cisplatin included cells with genetic lesions in the helicases REGULATOR OF TELOMERE ELONGATION HELICASE 1 (RTEL1, encoded by Cre02.g089608) (ref. [24]) and FANCONI ANEMIA COMPLEMENTATION GROUP M (FANCM, encoded by Cre03.g208833) and in the crossover junction endonuclease METHANSULFONATE UV SENSITIVE 81 (MUS81, encoded by Cre12.g555050).

Our data suggest several instances where a given factor is required for the repair of a specific class of DNA damage in *Chlamydomonas*, but not in *Arabidopsis*, or vice versa, suggesting lineage-specific differences in how DNA damage is repaired. For example, *Chlamydomonas fancm* mutants are sensitive to the DNA crosslinker cisplatin, whereas *Arabidopsis fancm* mutants are not[25]. Conversely, *Arabidopsis mus81* mutants are sensitive to the alkylating agent MMS and the DNA crosslinker mitomycin C[26], whereas *Chlamydomonas mus81* mutants were not.

Taken together, our data suggest that the core eukaryotic DNA repair machinery defined in other systems is generally conserved in green algae. Moreover, the observation of expected phenotypes illustrates the quality of the presented data and the utility of the platform for chemical genomic studies.

**Classification of genes based on photosynthesis phenotypes.** Our data allowed the classification of 38 genes whose disruption leads to a photoautotrophic growth defect into two clusters. One cluster consisted of genes whose disruption confers sensitivity to light when grown on medium supplemented with acetate, whereas the other contained genes whose disruption does not (Fig. 4b,c and Supplementary Data 2).

The light-sensitive cluster (Fig. 4c) included genes encoding core photosynthesis components and biogenesis factors such as the mRNA trans-splicing factors RNA MATURATION Of PSAA (RAA1)[27], RAA3[28], OCTOTRICOPEPTIDE REPEAT 120 (OPR120) and OPR104[29]; photosystem II biogenesis factor CONSERVED IN PLANT LINEAGE AND DIATOMS 10 (CPLD10)[29,30]; the chlorophyll biogenesis factor Mg-CHELATASE SUBUNIT D (CHLD)[31]; the ATP synthase translation factor TRANSLATION DEFICIENT ATPase 1 (TDA1)[32]; the Rubisco mRNA stabilization factor MATURATION OF RBCL 1 (MRL1)[33]; and the Calvin–Benson–Bassham cycle enzymes SEDOHEPTULOSE-BISPHOSPHATASE 1 (SEBP1)[34] and PHOSPHORIBULOKINASE 1 (PRK1)[35]. Several highly conserved but poorly characterized genes are also found in this cluster, including the putative Rubisco methyltransferase encoded by Cre12.g524500[36], the putative thioredoxin Cre01.g037800, the predicted protein with a domain of unknown function (DUF1995) Cre06.g281800 (which we named *LIGHT SENSITIVE*

*AND/OR ACETATE-REQUIRING 4* (*LSAR4*)) and Cre13.g572100 (which we named *LIGHT GROWTH SENSITIVE 4* (*LGS4*)), as well as four *Chlorophyta*-specific genes. The mutant phenotypes of these poorly characterized genes and their presence in this light-sensitive cluster together suggest that their products could mediate the biogenesis, function or regulation of core components of the photosynthetic machinery.

The low $CO_2$-sensitive cluster (Fig. 4b) contains known and new components of the algal $CO_2$-concentrating mechanism (CCM), as detailed below.

**New CCM components.** The CCM increases the $CO_2$ concentration around the $CO_2$-fixing enzyme Rubisco, thus enhancing the rate of carbon uptake. The mechanism uses carbonic anhydrases in the chloroplast stroma to convert $CO_2$ to $HCO_3^-$, which is transported into the lumen of the thylakoid membranes that traverse a Rubisco-containing structure called the pyrenoid[37]. There, the lower pH drives the conversion of $HCO_3^-$ back into concentrated $CO_2$ that feeds Rubisco[37]. Mutants deficient in the CCM are unable to grow photoautotrophically in air, but their photoautotrophic growth is rescued in 3% $CO_2$ (ref. [37]). We observed this phenotype for one or more alleles of genes whose disruption was previously shown to disrupt the CCM (Supplementary Table 4), including genes encoding the chloroplast envelope $HCO_3^-$ transporter LOW $CO_2$ INDUCIBLE GENE A (LCIA)[38], and the thylakoid lumen CARBONIC ANHYDRASE 3 (CAH3) (ref. [39]), the stromal carbonic anhydrase LOW $CO_2$ INDUCIBLE GENE B (LCIB)[40], the master transcriptional regulator CCM1/CIA5 (refs. [41,42]) and the pyrenoid structural protein STARCH GRANULES ABNORMAL 1 (SAGA1) (ref. [43]) (Supplementary Table 6).

We observed similar high $CO_2$ rescue of photoautotrophic growth defects for mutants in multiple poorly characterized genes in the light-insensitive cluster, suggesting that many of these genes are new components in the CCM. These genes formed a cluster with *SAGA1* (ref. [43]), the only previously known CCM gene with enough alleles to be present in the cluster. We named one of these genes, Cre06.g259100, *SAGA3* (*STARCH GRANULES ABNORMAL FAMILY MEMBER 3*) because its protein product shows homology to the two pyrenoid structural proteins SAGA1 and SAGA2 (ref. [44]) (Extended Data Fig. 4). Consistent with a role in the CCM, SAGA3 localizes to the pyrenoid[45]. We also observed this phenotype in mutants lacking the pyrenoid starch sheath-localized protein STARCH BRANCHING ENZYME 3 (SBE3) (ref. [46]), suggesting that this enzyme plays a key role in the biogenesis of the pyrenoid starch sheath, a structure surrounding the pyrenoid that was recently shown to be important for pyrenoid function under some conditions[47]. Our cluster also contains the gene encoding FUZZY ONIONS (FZO)-like (FZL), a dynamin-related membrane remodeling protein involved in thylakoid fusion and light stress; mutants in this gene have pyrenoid shape defects[48]. Our results suggest that thylakoid organization influences pyrenoid function. The cluster additionally includes genes encoding CLV1 (Cre13.g574000), a predicted voltage-gated chloride channel that we hypothesize is important for regulating the ion balance in support of the CCM or, alternatively, may directly mediate $HCO_3^-$ transport; a protein containing a Rubisco-binding motif[44] (Cre12.g528300, which we named LOW $CO_2$ SENSITIVE 1 (LOCO1)); and a predicted Ser-Thr kinase HIGH LEAF TEMPERATURE 1 (HT1) (Cre02.g111550). The kinase is a promising candidate for a regulator in the CCM, as multiple CCM components are known to be phosphorylated[49–51], but no kinase had previously been shown to have a CCM phenotype.

Also in this cluster are genes encoding the predicted PYRUVATE DEHYDROGENASE 2 (PDH2) (Cre03.g194200) and the predicted DIHYDROLIPOYL DEHYDROGENASE (DLD2) (Cre01.g016514). We hypothesize that these proteins are part of a glycine decarboxylase complex that functions in photorespiration,

a pathway that recovers carbon from the products of the Rubisco oxygenation reaction. PDH2 was found in the pyrenoid proteome[52], suggesting the intriguing possibility that glycine decarboxylation may be localized to the pyrenoid, where the released $CO_2$ could be recaptured by Rubisco.

**New genes with roles in cilia function.** *Chlamydomonas* cells swim using two motile cilia. To identify mutants with abnormal cilia function, we separated cells based on swimming ability by placing the mutant pool in a vertical column and collecting the supernatant and pellet. In this assay, mutants with altered swimming behavior were enriched in GO terms such as 'dynein complex', which comprises motor proteins involved in ciliary motility (Fig. 2e). Eighteen genes were represented by enough alleles to provide high confidence (FDR < 0.3) that their disruption produces a defect in swimming (Fig. 4d). These genes were enriched ($P = 0.0075$, Fisher's exact test) in genes encoding proteins found in the *Chlamydomonas* flagella proteome[53]. Half of these genes or their orthologs have previously been associated with a cilia-related phenotype in *Chlamydomonas* and/or mice (Supplementary Table 11).

In our analysis, these 18 genes formed four clusters that appeared to subclassify their function (Fig. 4d). The first cluster is enriched in known regulators of ciliary membrane composition and includes the gene encoding NEPHROCYSTIN-4-LIKE PROTEIN (NPHP4)[54]; the gene encoding its physical interactor TRANSMEMBRANE PROTEIN 67 (TMEM67, also named MECKEL SYNDROME TYPE 3 (MKS3) in mammals), which has been implicated in photoreceptor intraciliary transport[55]; and the gene encoding CENTRIOLE PROTEOME PROTEIN 290 (CEP290) (ref. [56]). We validated the swimming defect of *tmem67* and observed that the mutant has shorter cilia (Extended Data Fig. 5). The poorly annotated gene Cre15.g638551 clusters with these genes, suggesting that it may also regulate ciliary membrane composition.

The second cluster contains genes encoding BARDET-BIEDL SYNDROME 1 PROTEIN 1 (BBS1) and BBS9, components of the Bardet–Biedl syndrome-associated complex that regulates targeting of proteins to cilia[57]. The poorly annotated gene Cre15.g640502 clustered with these genes, suggesting that it may also play a role in targeting proteins to cilia.

The third cluster contains eight genes, four of which relate to the dynein complex, including the ciliary dynein assembly factor DYNEIN ASSEMBLY LEUCINE-RICH REPEAT PROTEIN (DAU1) (ref. [58,59]), OUTER DYNEIN ARM (ODA), DYNEIN ARM INTERMEDIATE CHAIN 1 (DIC1) (ref. [60]), DYNEIN HEAVY CHAIN 1 (DHC1) (ref. [61]) and TUBULIN-TYROSINE LIGASE 9 (TTLL9), which modulates ciliary beating through the addition of polyglutamate chains to alpha-tubulin[62]. The predicted thioredoxin peroxidase gene Cre04.g218750 and three poorly annotated genes (Cre07.g338850, Cre01.g012900 and Cre16.g675600) clustered with these genes, suggesting possible roles in dynein assembly or regulation.

The fourth cluster contains three poorly characterized genes, *FLAGELLA ASSOCIATED PROTEIN2* (*FAP2*), *FLAGELLA ASSOCIATED PROTEIN 81* (*FAP81*) and *TEF24*. The protein encoded by *FAP81* (Cre06.g296850) was identified in the *Chlamydomonas* cilia proteome[53], and its human homologue DELETED IN LUNG AND ESOPHAGEAL CANCER PROTEIN 1 (DLEC1) localizes to motile cilia[63]. We validated the swimming defect of the *fap81* mutant and established that it has shorter cilia (Extended Data Fig. 5). The localization to motile cilia in humans and our finding that mutating the encoding gene leads to a ciliary motility defect together suggest the intriguing possibility that impaired cilia motility contributes to certain lung and esophageal cancers.

**New genes required for actin cytoskeleton integrity.** Our analysis revealed a group of genes that render cells sensitive to LatB when

any are mutated (Fig. 4f). LatB binds to monomers of actin, one of the most abundant and conserved proteins in eukaryotic cells, and prevents actin polymerization[64] (Fig. 5a). LatB was first discovered as a small molecule that protects the sea sponge *Latrunculina magnifica* from predation by fish[65] and is an example of the chemical warfare that organisms use to defend themselves and compete in nature (Fig. 5b).

*Chlamydomonas* protects itself against LatB-mediated inhibition of its conventional actin INNER DYNEIN ARM5 (IDA5) by upregulating the highly divergent actin homologue NOVEL ACTIN-LIKE PROTEIN 1 (NAP1), which appears to perform most of the same functions as actin but is resistant to inhibition by LatB[66]. Upon inhibition of IDA5 by LatB, IDA5 is degraded and the divergent actin *NAP1* is expressed[66]. The expression of *NAP1* is dependent on three other known genes, *LatB-SENSITIVE1-3* (*LAT1–LAT3*) (Fig. 5c); thus, mutants lacking any of these four genes are highly sensitive to LatB[66].

Our phenotype data revealed three new components of this F-actin homeostasis pathway, which we named LAT5 (encoded by Cre17.g721950), LAT6 (encoded by Cre15.g640101) and LAT7 (encoded by Cre11.g482750). *LAT5* and *LAT6* clustered together with three previously known components of the pathway (*NAP1*, *LAT2* and *LAT3*), and disruption of all six genes rendered cells sensitive to LatB (Supplementary Table 6). Mutants in all three new components show a relatively mild phenotype when compared to those mutants in *LAT1–LAT3* (Fig. 5d), illustrating the sensitivity of our phenotyping platform.

Ubiquitin proteasome-mediated proteolysis of IDA5 has been hypothesized to drive the degradation of IDA5 and promote the formation of F-NAP1 (ref. [67]), but the factors involved were unknown. *LAT5* and *LAT6* encode predicted subunits of a SKP1, CDC53/CULLIN, F-BOX RECEPTOR (SCF) E3 ubiquitin ligase complex, whose homologues promote the degradation of target proteins[68]. The disruption of *LAT5* and *LAT6* impaired degradation of IDA5 upon LatB treatment, suggesting that LAT5 and LAT6 mediate IDA5 degradation (Fig. 5e). *LAT7* encodes a predicted importin, and its disruption impairs IDA5 degradation after LatB treatment (Fig. 5e), suggesting that nuclear import is required for IDA5 degradation.

It was previously not clear how broadly conserved this F-actin homeostasis pathway is. We found that the land plant model *Arabidopsis* has homologs of IDA5, NAP1, LAT3, LAT5, LAT6 and LAT7. We observed that *Arabidopsis* mutants disrupted in *LAT3*, *LAT5* and *LAT6* are sensitive to LatB treatment (Fig. 5f,g), which was not expected a priori, suggesting that this pathway for actin cytoskeleton integrity and the gene functions identified here are conserved in land plants.

## Discussion

In this work, we determined the phenotypes of 58,101 *Chlamydomonas* mutants across a broad variety of growth conditions. We observed a phenotype for mutants representing 10,380 genes, providing a valuable starting point for characterizing the functions of thousands of genes. Mutant phenotypes are searchable at chlamylibrary.org, and individual mutants can be ordered from the Chlamydomonas Resource Center.

We provided several examples of how the data enable discovery of gene functions and phenotypes in algae and plants. We validated our discovery of three new genes in the actin cytoskeleton integrity pathway, obtained insights into their molecular functions and found that this pathway appears to be conserved in land plants. We validated our discovery of cilia function defects for two new genes and our observation of an unexpected sensitivity of the chloroplast unfolded protein response to the alkylating agent MMS. We also discussed how our data provide insights and candidate genes in other pathways, including DNA damage repair, photosynthesis and the CCM.

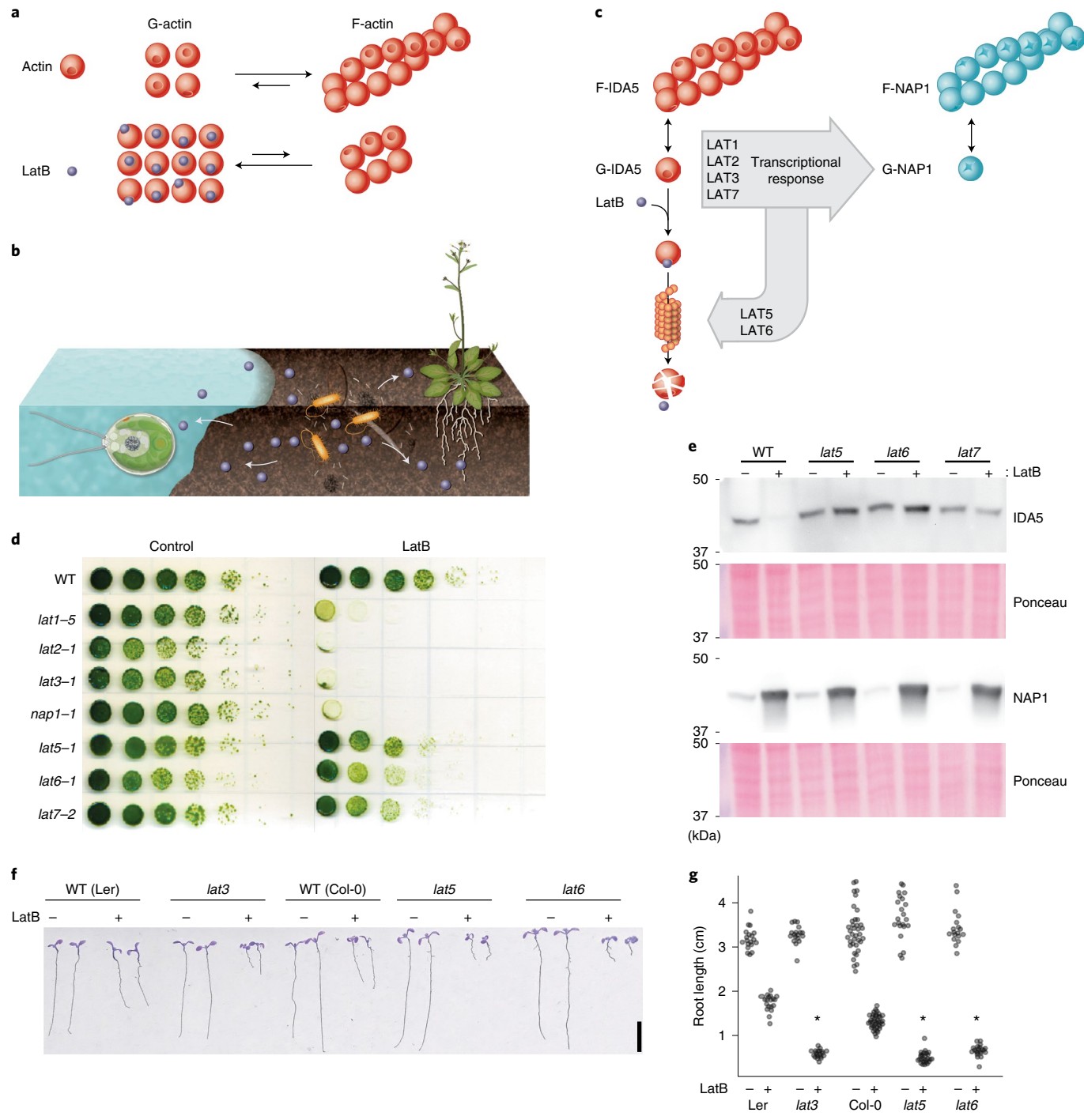

**Fig. 5 | The approach revealed new conserved components of a defense mechanism against cytoskeleton inhibitors. a**, LatB interferes with actin polymerization. **b**, Soil microorganisms deploy (arrows) actin inhibitors (blue circles) for a competitive advantage in their environment. **c**, *Chlamydomonas* responds to actin inhibition by degrading its conventional actin, IDA5, and upregulating an alternative actin, NAP1. **d**, Growth of new *lat* mutants identified in this study (*lat5-1, lat6-1* and *lat7-2*) was compared to previously isolated *lat1-5, lat2-1, lat3-1* and *nap1-1* mutants[66] in the absence (control) and presence (LatB) of 3 μM LatB. **e**, Immunoblot of conventional (IDA5) and alternative (NAP1) actins shows that *lat5-1, lat6-1* and *lat7-2* are deficient in actin degradation. Immunoblot representative of *n* = 3 independent experiments. **f**, The F-actin homeostasis pathway is conserved between green algae and plants. Mutants in *Arabidopsis* genes homologous to *Chlamydomonas lat3, lat5* and *lat6* are sensitive to LatB, as evidenced by decreased root length. **g**, Quantification of root length in *Arabidopsis* mutants. Asterisks mark significant changes relative to wild type under the same condition based on two-way analysis of variance. The exact value of $P = 2.4 \times 10^{-47}$ (Ler versus *lat3*), $P = 1.4 \times 10^{-62}$ (Col-0 versus *lat5*), $P = 6.8 \times 10^{-23}$ (Col-0 versus *lat6*). *n* = 26 roots examined over three independent experiments.

Altogether, 58% of the high-confidence gene–phenotype interactions involve a *Chlamydomonas* gene with a predicted *Arabidopsis* homologue (Supplementary Table 8); for approximately 79% of the corresponding *Arabidopsis* homologues, our data predict a new gene–phenotype relationship. This work illustrates the value of using a microbial photosynthetic organism for discovering gene

functions on a large scale. We hope that the genotype–phenotype relationships identified here will guide the characterization of thousands of genes, with potential applications in agriculture, the global carbon cycle and our basic understanding of cell biology.

## Online content

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

## Methods

**Library maintenance.** The *Chlamydomonas* mutant collection[12] was maintained by robotically passaging 384-colony arrays to fresh medium using a Singer RoToR robot (Singer Instruments, 704). The mutant collection was grown on 1.5% agar Tris-acetate-phosphate (TAP) medium with modified trace elements[69] in complete darkness at room temperature. The routine passaging interval of 4 weeks for library maintenance was shortened to 2 weeks during the time period of pooled screens to increase cell viability.

**Screening of LATCA to identify *Chlamydomonas* growth inhibitors.** LATCA[13] was used to identify molecules capable of inhibiting growth in wild-type *Chlamydomonas* (cMJ030). We found that 1,222 of these 3,650 LATCA compounds reduce growth by 90% at 25 μM (Supplementary Table 3). Due to resource limitations, we could not perform competitive growth experiments with all 1,222 active chemicals. Hence, we further selected the most active compounds and analyzed their structural similarity to identify the most diverse set of compounds for the competitive growth experiments. Dose–response experiments with 1,140 compounds validated activity for 954 compounds, and identified 136 chemicals that reduce growth at 2 μM or less (Supplementary Table 3). We then used the extended-connectivity fingerprint algorithm[70] to convert all LATCA compound structures into numerical fingerprints. Extended-connectivity fingerprints were then used to compute structural similarity of pairs of compounds on a scale of 0 to 1 using Tanimoto coefficients[71]. The set of Tanimoto coefficients between all pairs of inhibitors was visualized using Cytoscape[72]. We then used the most active inhibitors to further reduce the similarity network to 28 clusters of structures exhibiting high levels of biological activity and selected 52 of these chemicals for subsequent treatment of the *Chlamydomonas* mutant library (Extended Data Fig. 1, Supplementary Table 3 and Supplementary Data 1).

**Library pooling and competitive growth experiments.** The first two rounds of mutant library screening (R1 and R2) were performed with the entire mutant collection (550 384-mutant array plates) in 20-liter carboys (Supplementary Table 1 and Supplementary Table 2). Mutants were pooled from 5-day-old 384-colony array plates into liquid TAP medium at room temperature and low light. In R1, the pool included nine additional copies for each of three plates (668–670) in the collection to test how quantitatively we can track the relative abundance of mutants in the starting population. In R2, we pooled a subset of the mutant collection (plates 597–670) from 384-colony array plates and another subset from 1,536-colony array plates (101–596) to test the performance of denser colony arrays for pooled screens.

Subsequent rounds of mutant library screening (R3–R6) were performed on the rearrayed library (245 384-mutant array plates) in 2-liter bottles. Mutants were pooled from 5-day-old 1,536-colony array plates. Condensing the library from 384 to 1,536-colony array plates helped to both homogenize colony growth and reduce the laborious pooling procedure.

We produced subpools each containing cells from eight 384 or 1,535-colony array plates by using sterile glass spreaders to pool cells from the plates into 50-ml conical tubes containing 40 ml TAP medium. These subpools were mixed by pipetting to break cell clumps using a 10-ml serological pipette with a P200 tip attached to it. Then, all subpools were combined into the final mutant collection pool by pipetting the subpools through a 100-μm cell strainer (VWR, 10054-458). The final pool was mixed using a magnetic stir bar, and the cell density was measured (Invitrogen, Countess) and adjusted to $1 \times 10^5$ cells ml$^{-1}$. For experiments not performed in TAP medium, cells were pelleted ($1,000 \times g$, 5 min, room temperature) and washed twice with the actual medium used for the pooled growth experiment.

Aliquots of $2 \times 10^8$ cells were pelleted ($1,000 \times g$, 5 min, room temperature) by centrifugation and frozen to represent the relative abundance of each mutant in the starting population. These samples are denoted as 'initial'.

Cultures were inoculated with $2 \times 10^4$ cells ml$^{-1}$ in transparent 20-liter carboy tanks (R1 and R2) or standard 2-liter bottles (R3–R6) using aliquots of the final mutant pool. Cultures were inoculated with $2 \times 10^4$ cells ml$^{-1}$, and most experiments were performed in 2-liter vessels ($4 \times 10^7$ cells total) with ~58,000 mutants (Supplementary Tables 1 and 2), resulting in ~700 cells per mutant on average in the 2-liter competitive growth experiments. Cultures were grown under a broad variety of conditions (Supplementary Table 2) of which 49 had two or more replicates. Unless otherwise indicated, cells were grown in TAP medium with modified trace elements at pH 7.5 under constant light (100 μmol photons m$^{-2}$ s$^{-1}$ using Lumigrow Lumibar lights, catalog number 8100-5502; equal levels of red, blue and white light) at 22 °C, aerated with air and mixed using a conventional magnetic stirrer at 200 rpm. The cell density of competitive growth experiments was tracked, and aliquots of $2 \times 10^8$ cells were pelleted by centrifugation after approximately seven doublings, when the culture reached approximately $2 \times 10^6$ cells ml$^{-1}$. We sought to avoid letting the cultures reach stationary phase, where experiments are less reproducible. At seven divisions, the mutant pool was typically in the late exponential growth phase. Cell pellets were frozen for subsequent DNA extraction and barcode quantification. Algal predator experiments were performed with *Daphnia magna*, *Philodina* sp. (Rotifer) and *Hypsibius exemplaris* (Tardigrade) purchased from Carolina Biological Supply.

**DNA extraction.** Total genomic DNA was extracted from frozen cell pellets representing $2 \times 10^8$ cells of each sample (initial, control and treatment).

First, frozen pellets were thawed at room temperature and resuspended in 1.6 ml resuspension buffer (1% SDS, 200 mM NaCl, 20 mM EDTA and 50 mM Tris-HCl, pH 8.0).

Second, 2 ml phenol/chloroform/isoamyl alcohol (25:24:1) was added to each sample and mixed by vortexing. This solution was then transferred into 15-ml Qiagen MaXtract High Density tubes (catalog number 129065) and centrifuged at 3,500 rpm for 5 min. Subsequently, the aqueous phase was transferred to a new 15-ml conical tube, 6.4 μl RNase A was added and the solution was incubated at 37 °C for 30 min. The phenol/chloroform/isoamyl alcohol extraction was then repeated, and the aqueous phase was transferred into a new 15-ml Qiagen MaXtract High Density tube before adding 2 ml phenol/chloroform/isoamyl alcohol (25:24:1). This solution was mixed by vortexing and centrifuged at 3,500 rpm for 5 min; then, 400-μl aliquots of the aqueous phase were transferred to 1.5-ml reaction tubes for DNA precipitation (typically four aliquots per sample).

Third, 1 ml ice-cold 100% ethanol was added to the solution to precipitate DNA. The tubes were gently mixed and incubated at –20 °C overnight. The DNA was pelleted at 13,200 rpm and 4 °C. The supernatant was discarded and the pellet washed in 1 ml 70% ethanol. The supernatant was discarded again, and the pellet was air-dried before resuspension in 50 μl water. Subsequently, the elution fractions of each sample were pooled and the DNA concentration was measured using a Qubit fluorometer (Invitrogen).

**Internal barcode amplification and Illumina library preparation.** Internal barcodes were amplified using Phusion Hot Start II (HSII) DNA Polymerase (Thermo Fisher, F549L) using previously described primers[12].

The 50-μl PCR mixture for 5′ barcode amplification contained 125 ng genomic DNA, 10 μl GC buffer, 5 μl DMSO, 1 μl dNTPs at 10 mM, 1 μl MgCl$_2$ at 50 mM, 2.5 μl of each primer at 10 μM and 1 μl Phusion HSII polymerase. Eight tubes of the PCR mixture were processed per sample and incubated at 98 °C for 3 min, followed by 10 three-step cycles (98 °C for 10 s, 58 °C for 25 s and 72 °C for 15 s) and then 11 two-step cycles (98 °C for 10 s and 72 °C for 40 s).

The 50-μl PCR mixture for 3′ barcode amplification contained 125 ng genomic DNA, 10 μl GC buffer, 5 μl DMSO, 1 μl dNTPs at 10 mM, 2 μl MgCl$_2$ at 50 mM, 2.5 μl of each primer at 10 μM and 1 μl Phusion HSII polymerase. Eight tubes of the PCR mixture were processed per sample and incubated at 98 °C for 3 min, followed by 10 three-step cycles (98 °C for 10 s, 63 °C for 25 s and 72 °C for 15 s) and then 11 two-step cycles (98 °C for 10 s and 72 °C for 40 s).

The PCR products of each sample were pooled for further processing. First, successful PCR was confirmed on a TBE 8% agarose gel in 1x Tris Borate EDTA before concentrating the PCR products on a Qiagen MinElute column and measuring the DNA concentration on a Qubit fluorometer. Second, 200–250 ng of up to 16 3′ or 5′ PCR products were combined into an Illumina HiSeq2000 library. Third, the internal barcode bands of the Illumina HiSeq2000 libraries were gel-purified and subjected to quality control on an Agilent Bioanalyzer. In addition, DNA concentration was determined on a Qubit fluorometer. Fourth, HiSeq2000 libraries were sequenced at the Genome Sequencing Service Center at Stanford University (Palo Alto, CA).

**Data analysis.** Initial reads were trimmed using cutadapt version 1.7.1 (ref. [73]) using the command 'cutadapt -a <seq> -e 0.1 -m 21 -M 23 input_file.gz -o output_file.fastq', where <seq> is GGCAAGCTAGAGA for 5′ data and TAGCGCGGGGCGT for 3′ data. Barcodes were counted by collapsing identical sequences using 'fastx_collapser' (http://hannonlab.cshl.edu/fastx_toolkit) and denoted as '_read_count'. Across all experiments conducted, ~62 million barcode read counts were determined. Barcode read counts for each dataset were normalized to a total of 100 million and denoted as '_normalized_reads' (Supplementary Table 12). Replicate control treatments performed in the same screening round were averaged by taking the mean of the normalized read counts to generate the average normalized read count (denoted as '_average_normalized_reads'). To calculate a '_read_count' (nonnormalized) for the averaged samples, the read counts for all of the averaged samples were summed and denoted as the '_average_read_count'. Control treatments that were averaged are denoted with 'average' and can be found in Supplementary Table 13.

Mutants in the library contain on average 1.2 insertions[12], each of which may contain a 5′ barcode, a 3′ barcode, both barcodes or potentially more than two barcodes if multiple cassettes were inserted at the loci. To represent a given insertion within a mutant, we selected a single barcode to represent it. All barcodes associated with the same gene and deconvoluted to the same library well and plate position were assumed to be from the same insertion and were then compared to identify the barcode with the highest read counts in the initial samples (R2–R6) to serve as the representative barcode.

To identify mutants with growth defects or enhancements due to a specific treatment, we compared the abundance of each mutant after growth under the treatment condition to its abundance after growth under a control condition. We called this comparison a 'screen' and the ratio of these abundances the 'mutant phenotype'. In order for a phenotype to be calculated, we required the

control treatment to have a read count above 50, which allowed for 16.8 million phenotypes to be determined.

To identify high-confidence gene–phenotype relationships we developed a statistical framework that leverages multiple independent mutant alleles. For each gene, we generated a contingency table of the phenotypes, Φ, by counting the number of alleles that met the following thresholds: [Φ < 0.0625, 0.0625 ≤ Φ < 0.125, 0.125 ≤ Φ < 0.25, 0.25 ≤ Φ < 0.5, 0.5 ≤ Φ < 2.0, 2.0 ≤ Φ < 4.0, 4.0 ≤ Φ < 8.0, 8.0 ≤ Φ < 16.0]. Only alleles that were mapped with confidence level 4 or less (corresponding to a likelihood of correct mapping of 58% or higher)[12] had an insertion in CDS/intron/5′ UTR feature, and had greater than 50 reads in the control condition were included in the analysis. The frequency of cassette insertion location based on gene feature was intron 25%, 3′ UTR 23%, CDS 19%, not mapped 14%, intergenic 6%, 5′ UTR 5% and multiple or others 8%. The insertion cassette used to generate the mutants contains two transcriptional terminators; thus, we reasoned that insertions in 5′ UTRs, introns and exons will lead to transcriptional disruption and loss-of-function mutants. Mutants with cassette integrations in the 3′ UTR were not expected to result in transcriptional disruption so were excluded from our statistical framework. A P value was generated for each gene by using Fisher's exact test to compare a gene's phenotype contingency table to a phenotype contingency table for all insertions in the screen. An FDR was performed on the P values of genes with more than two alleles using the Benjamini–Hochberg method[74].

To determine a representative phenotype for a gene, the median phenotype for all alleles of that gene that were included in the Fisher's exact test was used. For clustering, these gene phenotypes were normalized by setting the median value of all gene phenotypes in a screen to zero. Clustering was performed with Python (2.7.11) packages SciPy (0.17.0) (ref. [75]) and visualized with Seaborn (0.7.1). To generate the hierarchical cluster in Fig. 4a, the pairwise distance was calculated using the 'correlation' metric, which calculates the correlation (Pearson) distance. The linkage matrix was calculated using the 'average' method. Pairwise Pearson correlation coefficients between gene phenotypes (Extended Data Fig. 3 and Supplementary Table 10) were calculated in Pandas (0.18.1). Transcriptome correlation data was collected, curated and analyzed in the Merchant laboratory[16]. Data were plotted and visualized with the Python packages Matplotlib (1.5.1) and Seaborn (0.7.1).

To determine if biological functions were associated with specific screens, we performed a GO term enrichment analysis. Using the same approach as with genes, we generated contingency tables of mutant phenotypes for each GO term. If a mutant's insertion is within a gene that had multiple GO term annotations, the mutant's phenotype data was added to each GO term's contingency table. A P value was generated for each GO term by using Fisher's exact test to compare a GO term's phenotype contingency table to a phenotype contingency table for all GO terms in the screen. An FDR was performed on the P values using the Benjamini–Hochberg method[74]. Clustering was performed (Extended Data Fig. 2) in Seaborn using the 'Euclidean' metric to calculate the pairwise distance and the 'average' method to calculate the linkage.

All analysis was performed using JGI Phytozome release v5.0 of the *Chlamydomonas* assembly and v5.6 of the *Chlamydomonas* annotation[76]. Gene identifiers (CreXX.gXXXXXX) can be used to link data found in the supplemental tables to gene annotation updates. All data have been deposited in Dryad Digital Repository (https://doi.org/10.6086/D1Q96Z). Custom code used for data analysis has been deposited in Zenodo (https://doi.org/10.5281/zenodo.6340170) (ref. [77]).

**MMS growth assays and VIPP2 immunoblot analysis.** The following strains were used[15]: WT = CC-4533; mars1 = mars1-3; mars1:MARS1-D = mars1-3 transformed with the MARS1-D transgene containing a 3×-Flag epitope after Met139; and mars1:MARS1-D KD = mars1-3 transformed with a catalytically-inactive MARS1-D bearing the kinase active site D1871A mutation. Before starting liquid cultures in TAP medium, all strains were restreaked in fresh TAP plates and grown in similar light conditions (i.e., ~50–70 μmol photons m$^{-2}$ s$^{-1}$, ~22 °C) for about 5–6 days. Before starting the MMS treatment, all strains were preconditioned in liquid cultures for 3 or 4 days. Next, cell cultures were equally diluted to ~5 μg chlorophyll ml$^{-1}$ and incubated in the presence or absence of MMS for 48 h. A 1% (vol/vol) MMS stock solution (Sigma-Aldrich, 129925) was freshly prepared in double-distilled H$_2$O at the beginning of each experiment. This MMS stock solution was further diluted 200 times directly into TAP medium to a final concentration of 0.05% (vol/vol). All chlorophyll concentration measurements were performed using a previously described methanol extraction method[78].

VIPP2 and alpha-TUBULIN immunoblot analyses were carried out as described previously[15] using denatured total protein samples prepared from liquid cultures incubated for 27 h in the presence or absence of 0.05% (vol/vol) MMS.

**Cilia-related mutant phenotyping.** Cilia mutants were grown in liquid TAP medium until they reached exponential phase. Cells were then mounted in u-Slide 8-well chambers (Ibidi, 80826) with 2% low-melting-point agarose (Sigma-Aldrich, A9414). Cilia defects were scored using a Leica DMi8 inverted microscope. Cilia length was measured using Fiji. Cilia swimming behavior was scored using TAP agar plates with 0.15% agar.

**Chlamydomonas lat mutant phenotyping.** Mutants used in this study are listed in Supplementary Table 14, and sequence information for all genotyping primers is summarized in Supplementary Table 15. Individual mutants were grown with gentle agitation at 100 μmol photons m$^{-2}$ s$^{-1}$. Disruption of LAT5, LAT6 and LAT7 genes (Cre17.g721950, Cre15.g640101 and Cre11.g482750) in the original isolates of lat5-1, lat6-1 and lat7-2 were confirmed by PCR. These mutants were then backcrossed with CC-124 or CC-125 three times, with perfect linkage of paromomycin resistance and LatB sensitivity in at least 10 tetrads observed after each round. The backcrossed strains and the previously established lat1-5, lat2-1, lat3-1 and nap1-1 mutants in the CC-124 background[66] were spotted on TAP agar containing 0.1% DMSO with or without 3 μM LatB (Adipogen, AG-CN2-0031, lot A00143/J) as 5× serial dilutions.

**Immunoblot materials IDA5 and NAP1 immunoblot analyses.** Cells were grown in liquid TAP medium at 21 °C with gentle agitation under 100 μmol photons m$^{-2}$ s$^{-1}$ and collected by centrifugation. Pellets were frozen in liquid nitrogen and subsequently resuspended in 100 μl ice-cold PNE buffer (10 mM phosphate, pH 7.0, 150 mM NaCl$_2$ and 2 mM EDTA) supplemented with 2× concentration of complete protease-inhibitor cocktail (Roche, 11697498001) and disrupted by vortexing with acid-washed glass beads. These samples were mixed directly with SDS–PAGE sample buffer, boiled for 3 min and cleared of debris by centrifugation at 12,000× g for 10 min at 4 °C before electrophoresis. SDS–PAGE was performed using 11% Tris-glycine. Blots were stained using a mouse monoclonal anti-actin antibody (EMD Millipore, clone C4, MAB1501), which recognizes IDA5, but not NAP1, and a rabbit anti-NAP1 antibody (generous gift from R. Kamiya and T. Kato-Minoura), which recognizes NAP1, but not IDA5. Horseradish peroxidase-conjugated anti-mouse IgG (ICN Pharmaceuticals, 55564) or anti-rabbit IgG (Southern Biotech, 4050-05) were used as secondary antibodies, respectively.

**Arabidopsis lat mutant phenotyping.** Mutants used in this study are listed in Supplementary Table 14 and sequence information for all genotyping primers is summarized in Supplementary Table 15. Seeds were surface-sterilized in 20% bleach for 5 min. Seeds were then rinsed with sterile water four times and stored at 4 °C for 3 days in the dark. After stratification, seeds were sown into square 10 cm × 10 cm petri plates containing full-strength Murashige and Skoog medium (MSP01-50LT), 1% agar (Duchefa, 9002-18-0), 1% sucrose and 0.05% MES and adjusted to pH 5.7 with 1 M KOH. Seedlings were grown in the presence of LatB (Sigma, L5288) or mock control containing an equivalent volume of the LatB solvent, DMSO. Plates were imaged using a CanonScan 9000 flatbed scanner. Root lengths were quantified using Fiji. Two-way analysis of variance and data visualization were done using Python.

**Reporting Summary.** Further information on research design is available in the Nature Research Reporting Summary linked to this article.

## Data availability
Mutant barcode read count data (Supplementary Table 12) and mutant phenotypes across all screens (Supplementary Table 4) can be found at https://doi.org/10.6086/D1Q96Z in the Dryad Digital Repository. Source data are provided with this paper.

## Code availability
Custom code used for data analysis has been deposited in Zenodo (https://doi.org/10.5281/zenodo.6340170)[77]. Source data are provided with this paper.

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

## Acknowledgements

We thank M. Cahn for developing and improving the CLiP website; X. Ji at the Stanford Functional Genomics Facility and Z. Weng at the Stanford Center for Genomics and Personalized Medicine for deep-sequencing services; K. Barton, W. Briggs and Z.-Y. Wang for providing lab space; T. Raab for providing lab equipment; J. Robertson and N. Ivanova for helping with library maintenance; M. Sison-Mangus for advice on *Daphnia* experiments; T. Kato-Minoura and R. Kamiya for the anti-NAP1 antibody; J. R. Pringle for supporting M.O.; J. Whitney for assistance in figure preparation; and members of the Dinneny, Jonikas and Jinkerson Labs and T. Xiang for constructive suggestions on the manuscript. This project was supported by grants from the National Institutes of Health (DP2-GM-119137), the National Science Foundation (MCB-1914989 and MCB-1146621) and the Simons Foundation and Howard Hughes Medical Institute (55108535) awarded to M.C.J.; a German Academic Exchange Service (DAAD) research fellowship awarded to F.F.; Simons Foundation fellowships of the Life Sciences Research Foundation awarded to R.E.J. and J.V.-B.; an EMBO long-term fellowship (ALTF 1450-2014 and ALTF 563-2013) awarded to J.V.-B. and S.R.; an National Science Foundation MCB Grant (1818383) awarded to M.O.; a Swiss National Science Foundation Advanced PostDoc Mobility Fellowship (P2GEP3_148531) awarded to S.R.; and a Simons Foundation and Howard Hughes Medical Institute (55108515) Faculty Scholars grant awarded to J.R.D. S.W. was supported by the U.S. Department of Energy, Office of Science, Basic Energy Sciences, Chemical Sciences, Geosciences, and Biosciences Division under field work proposal 449B. K.K.N. is an investigator at the Howard Hughes Medical Institute. Work in the Merchant laboratory is supported by a cooperative agreement with the US Department of Energy Office of Science, Office of Biological and Environmental Research program under award DE-FC02-02ER63421.

## Author contributions

S.W. and K.K.N. performed rose bengal treatments; R.G.K., Y.K. and A.R.G. performed anoxia and high light treatments; J.V.-B., F.F. and R.E.J. prepared mutant pools, performed all other treatments and processed all samples; S.R.C. provided the LATCA compounds; M.M., J.O., C.P. and M.N. validated the active LATCA compounds; M.M. performed chemoinformatics analysis; M.O. validated LatB phenotypes in *Chlamydomonas* and performed immunoblots; S.R. and P.W. validated the *mars1* phenotype; W.P. guided statistical analysis and website development; P.A.S. and S.S.M. provided transcriptomics data; X.L. provided early access to the *Chlamydomonas* mutant library; J.V.-B. and J.R.D. confirmed *Arabidopsis* phenotypes; J.V.-B., F.F., M.C.J. and R.E.J. designed experiments and analyzed and interpreted the data; and J.V.-B., F.F., M.C.J. and R.E.J. wrote the manuscript, with input from all authors.

## Competing interests

J.V.-B., F.F., M.C.J. and R.E.J. note that a provisional patent application (US 63/123,422) on aspects of these findings has been submitted to the USPTO. The other authors declare no competing interests.

## Additional information

**Extended data** is available for this paper at https://doi.org/10.1038/s41588-022-01052-9.

**Correspondence and requests for materials** should be addressed to José R. Dinneny, Martin C. Jonikas or Robert E. Jinkerson.

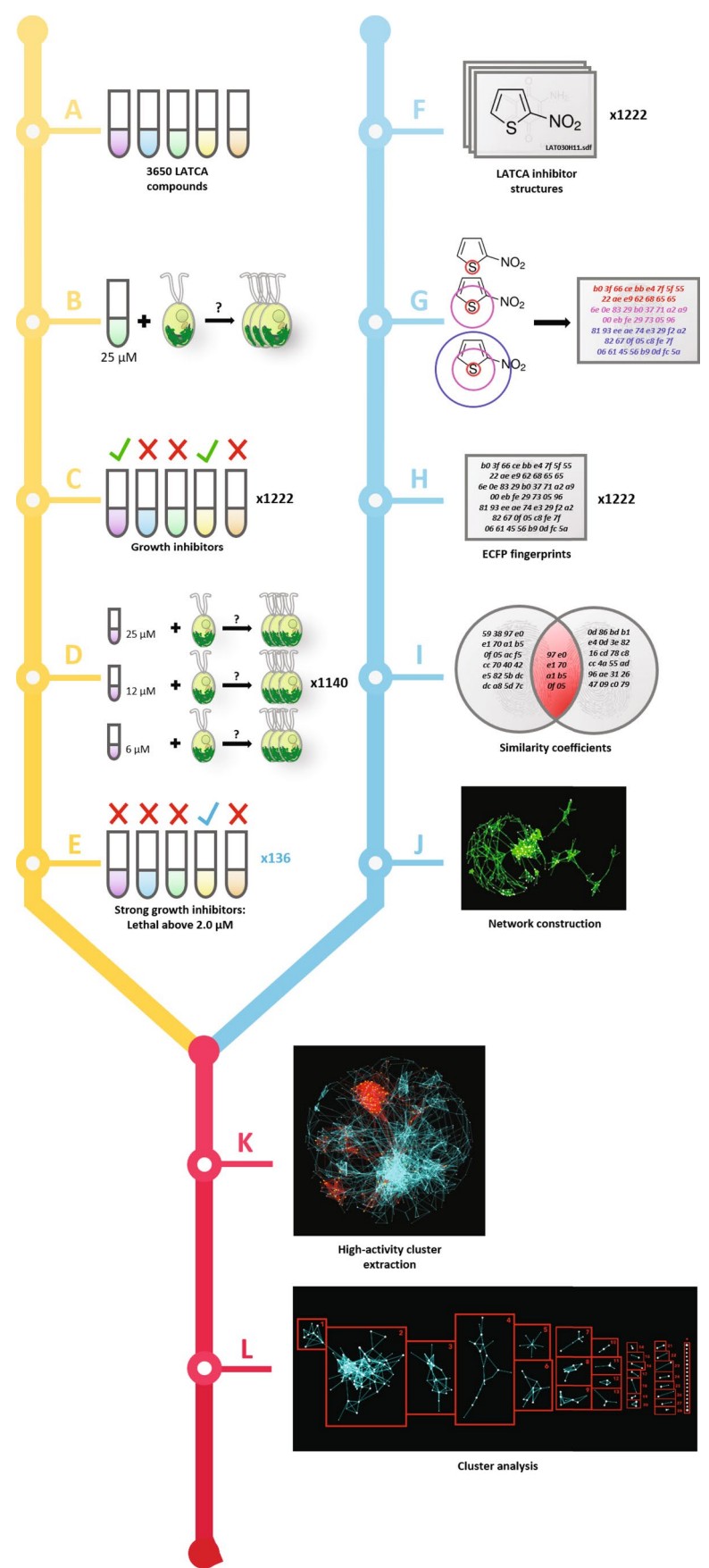

**Extended Data Fig. 1 | See next page for caption.**

**Extended Data Fig. 1 | A screen of the chemical library 'LATCA' identified 1,222 inhibitors of Chlamydomonas growth, 136 of which are active at 2 µM or less.** Phase 1 of the LATCA screen is depicted in A–E: The growth rate of wild-type *Chlamydomonas* (cMJ030) was evaluated in TAP and TP in the presence of 3,650 LATCA compounds. 1,222 out of the 3,650 LATCA compounds reduced growth by 90% or more at 25 µM (**a–c**). Dose-response experiments were performed in TAP medium with 1,140 out of the 1,222 highly active compounds that reduced growth at 2 µM or less (**d,e**). Phase 2 of the LATCA screen is depicted in F–J: Structural data files (SDFs) were acquired for all LATCA inhibitors (**f**) and converted into numerical fingerprints (extended-connectivity fingerprints; ECFPs) (**g, h**). ECFPs were then used to compute the structural similarity of pairs of compounds using Tanimoto coefficients (**i**). The set of Tanimoto coefficients between all pairs of inhibitors was condensed into a usable network (**j**). Phase 3 of the LATCA screen is depicted in K and L: Data from A–E was used to further reduce the similarity network from J to 28 clusters of structures exhibiting high levels of growth inhibition along with a group of singleton structures (*) that did not cluster. Supplementary Table 3 summarizes data A–E and shows cluster annotations from L; see also Extended Data File 1 for all chemical structures from L.

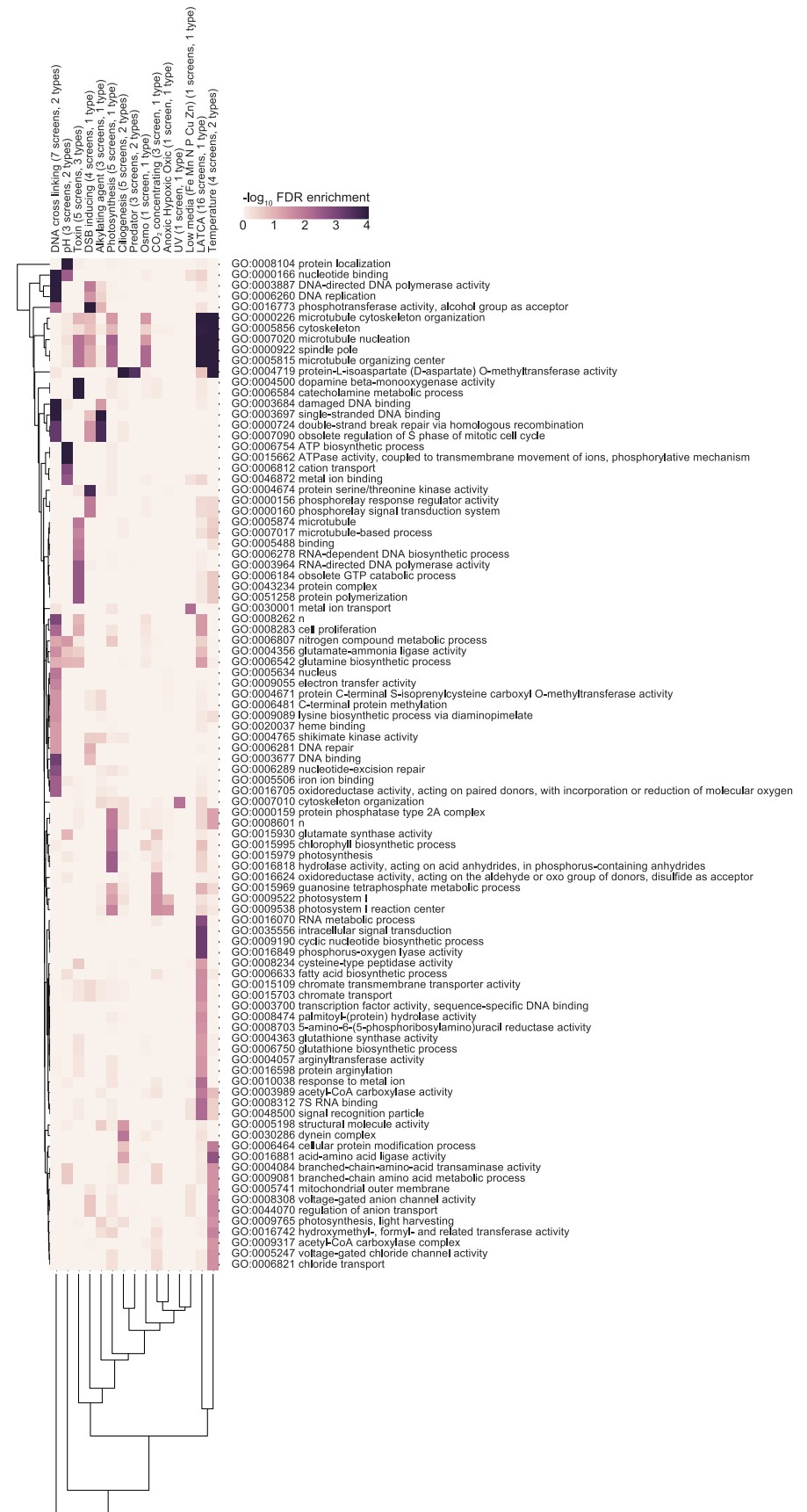

**Extended Data Fig. 2 | Full GO term enrichments.** Gene Ontology term analysis reveals enrichment of biological functions observed for specific screens. GO; FDR < 0.05.

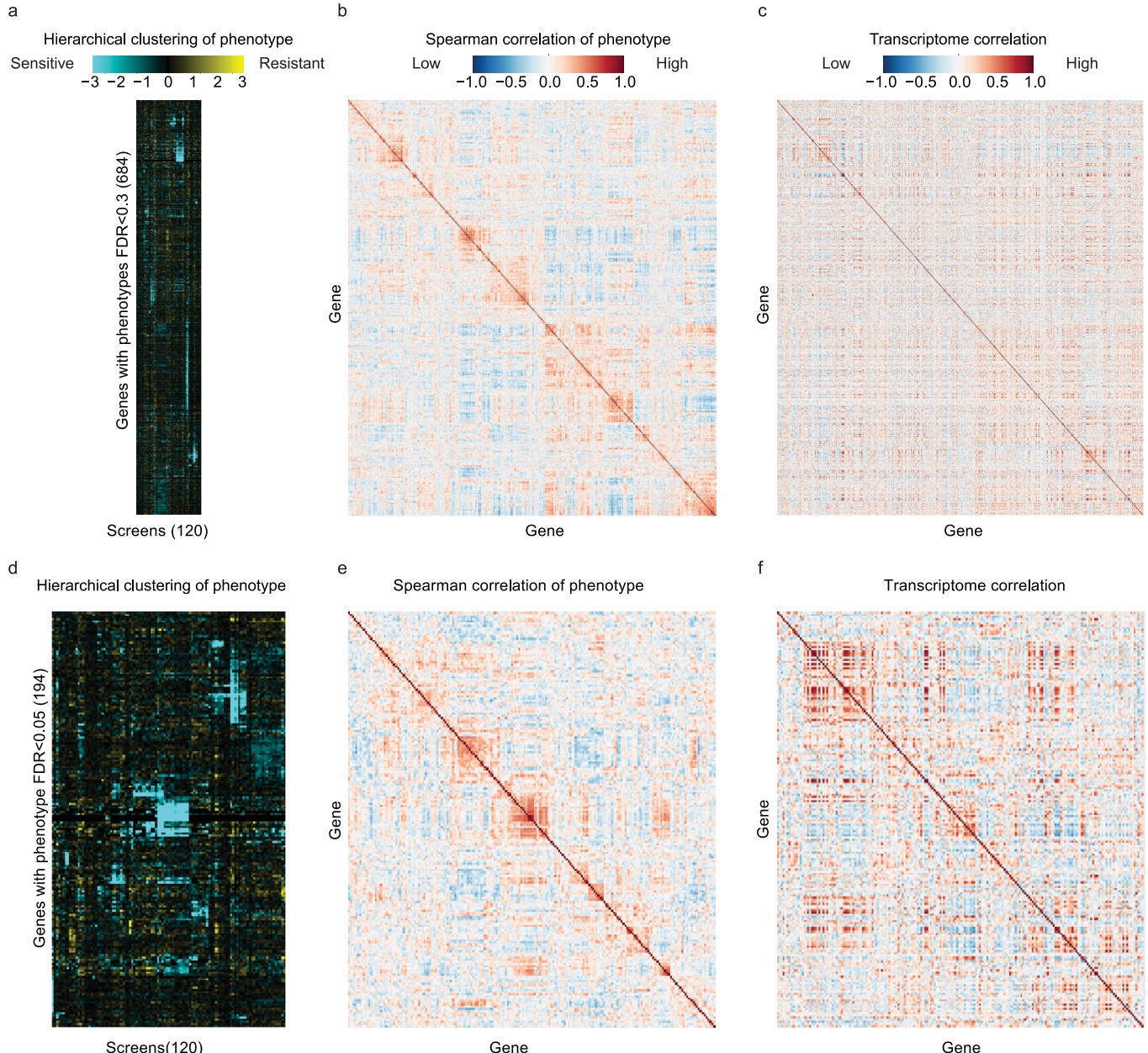

**Extended Data Fig. 3 | Comparison of phenotypic and transcriptomic correlations. a,** 684 genes were clustered based on the similarity of their phenotypes across 120 screens. **b,** Spearman correlation matrix of phenotypes (FDR < 0.3). **c,** Transcriptome correlation of gene with phenotype (FDR < 0.3). **d,** 194 genes were clustered based on the similarity of their phenotype across 120 screens. **e,** Spearman correlation matrix of phenotypes (FDR < 0.05). **f,** Transcriptome correlation of gene with phenotype (FDR < 0.05). Data can be found in Supplementary Table 10.

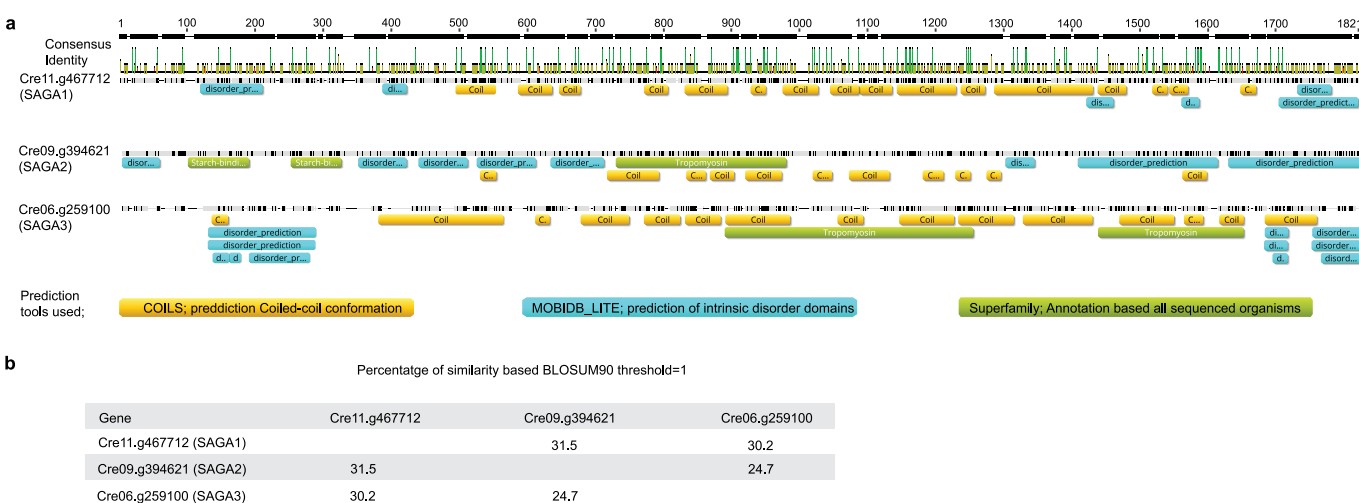

**Extended Data Fig. 4 | SAGA protein alignments. a**, Alignments of SAGA1, SAGA2, and SAGA3. Domain annotation was based on three different tools under the Geneious visualization platform. **b**, BLOSUM90 alignments between SAGA proteins.

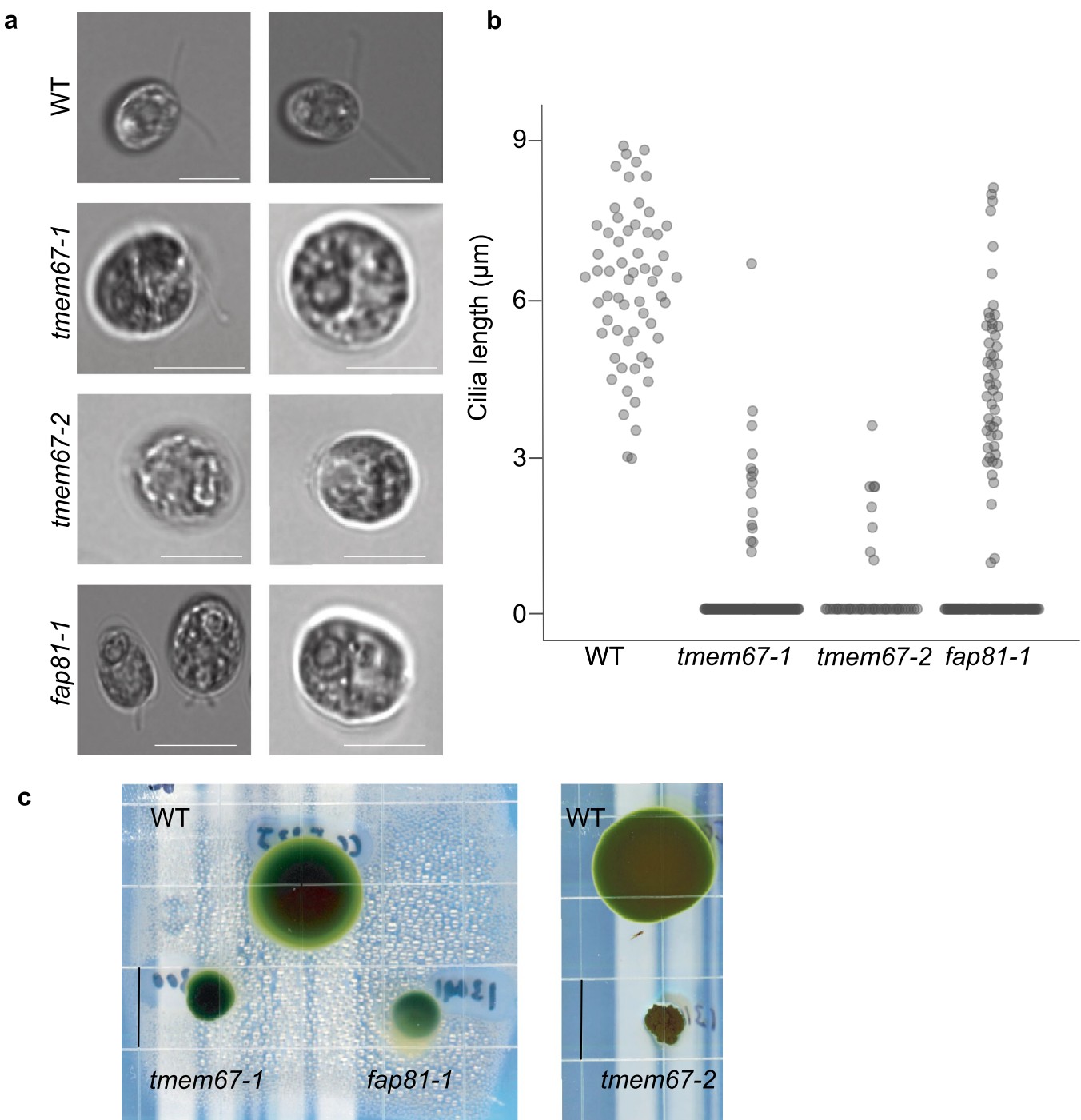

**Extended Data Fig. 5 | Validation of cilia mutant phenotypes. a**, Bright-field microscopy microscope images of cilia mutants show defects in ciliary length. Scale bar: 10 μm. **b**, Quantification of cilia length. n =number of samples (WT n = 60, *tmem67-1* n = 112, *tmem67-2* n = 46, *fap81-1* n = 139). **c**, Swimming behavior of mutants, as determined by growth on TAP medium solidified with 0.15% agar. Scale bars: 5 cm.

# nature research

# Reporting Summary

Nature Research wishes to improve the reproducibility of the work that we publish. This form provides structure for consistency and transparency in reporting. For further information on Nature Research policies, see our Editorial Policies and the Editorial Policy Checklist.

## Statistics

For all statistical analyses, confirm that the following items are present in the figure legend, table legend, main text, or Methods section.

| n/a | Confirmed | |
|---|---|---|
| ☐ | ☒ | The exact sample size (*n*) for each experimental group/condition, given as a discrete number and unit of measurement |
| ☐ | ☒ | A statement on whether measurements were taken from distinct samples or whether the same sample was measured repeatedly |
| ☐ | ☒ | The statistical test(s) used AND whether they are one- or two-sided *Only common tests should be described solely by name; describe more complex techniques in the Methods section.* |
| ☒ | ☐ | A description of all covariates tested |
| ☐ | ☒ | A description of any assumptions or corrections, such as tests of normality and adjustment for multiple comparisons |
| ☐ | ☒ | A full description of the statistical parameters including central tendency (e.g. means) or other basic estimates (e.g. regression coefficient) AND variation (e.g. standard deviation) or associated estimates of uncertainty (e.g. confidence intervals) |
| ☐ | ☒ | For null hypothesis testing, the test statistic (e.g. *F*, *t*, *r*) with confidence intervals, effect sizes, degrees of freedom and *P* value noted *Give P values as exact values whenever suitable.* |
| ☒ | ☐ | For Bayesian analysis, information on the choice of priors and Markov chain Monte Carlo settings |
| ☐ | ☒ | For hierarchical and complex designs, identification of the appropriate level for tests and full reporting of outcomes |
| ☐ | ☒ | Estimates of effect sizes (e.g. Cohen's *d*, Pearson's *r*), indicating how they were calculated |

*Our web collection on statistics for biologists contains articles on many of the points above.*

## Software and code

Policy information about availability of computer code

| Data collection | No software was used for data collection. |
|---|---|
| Data analysis | Analysis was conducted with Python (2.7.11), SciPy (0.17.0), Pandas (0.18.1), Matplotlib (1.5.1), and Seaborn (0.7.1). Details can be found in the methods. Custom code used for data analysis has been deposited in Zenodo (https://doi.org/10.5281/zenodo.6340170). |

For manuscripts utilizing custom algorithms or software that are central to the research but not yet described in published literature, software must be made available to editors and reviewers. We strongly encourage code deposition in a community repository (e.g. GitHub). See the Nature Research guidelines for submitting code & software for further information.

## Data

Policy information about availability of data

All manuscripts must include a data availability statement. This statement should provide the following information, where applicable:
- Accession codes, unique identifiers, or web links for publicly available datasets
- A list of figures that have associated raw data
- A description of any restrictions on data availability

All data, including mutant barcode read count data (Supplementary Table 12) and mutant phenotypes across all screens (Supplementary Table 4), can be found at https://doi.org/10.6086/D1Q96Z in the Dryad Digital Repository.

# Field-specific reporting

Please select the one below that is the best fit for your research. If you are not sure, read the appropriate sections before making your selection.

☒ Life sciences ☐ Behavioural & social sciences ☐ Ecological, evolutionary & environmental sciences

For a reference copy of the document with all sections, see nature.com/documents/nr-reporting-summary-flat.pdf

# Life sciences study design

All studies must disclose on these points even when the disclosure is negative.

| | |
|---|---|
| Sample size | We empirically determined the sample sizes based on published research. |
| Data exclusions | Data from screens where the conditions were extremely harsh, such as camptothecin, that resulted in population bottlenecks where the majority of mutants/barcodes are lost were excluded from analysis. |
| Replication | Replicates of many screens were performed. Most mutants have similar phenotypes in screen replicates (Fig. 2C). |
| Randomization | Randomization was not applicable for this study. |
| Blinding | Blinding and randomization were not used for this study. |

# Reporting for specific materials, systems and methods

We require information from authors about some types of materials, experimental systems and methods used in many studies. Here, indicate whether each material, system or method listed is relevant to your study. If you are not sure if a list item applies to your research, read the appropriate section before selecting a response.

## Materials & experimental systems

| n/a | Involved in the study |
|---|---|
| ☐ | ☒ Antibodies |
| ☐ | ☒ Eukaryotic cell lines |
| ☒ | ☐ Palaeontology and archaeology |
| ☐ | ☒ Animals and other organisms |
| ☒ | ☐ Human research participants |
| ☒ | ☐ Clinical data |
| ☒ | ☐ Dual use research of concern |

## Methods

| n/a | Involved in the study |
|---|---|
| ☒ | ☐ ChIP-seq |
| ☒ | ☐ Flow cytometry |
| ☒ | ☐ MRI-based neuroimaging |

## Antibodies

| | |
|---|---|
| Antibodies used | anti-actin antibody (clone C4, EMD Millipore, MAB1501); HRP-conjugated anti-mouse IgG (ICN Pharmaceuticals; 55564); anti-rabbit IgG (Southern Biotech; 4050-05); rabbit anti-NAP1 antibody (generous gift from Ritsu Kamiya and Takako Kato-Minoura PMID:12796293). The following dilutions were used: Anti-actin, 1:500; Anti-mouse HRP, 1:5000; Anti-NAP1, 1:1000; Anti-rabbit HRP, 1:20,000. |
| Validation | We observed decreasing abundance of IDA5 upon addition of LatB, conversely we observed increased abundance of NAP1 consistent with these antibodies recognizing their intended targets. |

## Eukaryotic cell lines

Policy information about cell lines

| | |
|---|---|
| Cell line source(s) | CC-4533 CW15 MT- [JONIKAS CMJ030] www.chlamycollection.org |
| Authentication | N/A |
| Mycoplasma contamination | N/A |
| Commonly misidentified lines (See ICLAC register) | N/A |

# Animals and other organisms

Policy information about <u>studies involving animals</u>; <u>ARRIVE guidelines</u> recommended for reporting animal research

| | |
|---|---|
| Laboratory animals | Predators of algae were purchased from Carolina Biologic and cared for based on their recommendations. |
| Wild animals | N/A |
| Field-collected samples | N/A |
| Ethics oversight | No institutional ethical approval was required because the animals were invertebrates (Daphnia, rotifers, and water bears) |

Note that full information on the approval of the study protocol must also be provided in the manuscript.

