## [Peer Review File · Nature Genetics]

Peer Review Information

Manuscript Title: Systematic characterization of gene function in the photosynthetic alga *Chlamydomonas reinhardtii*

Corresponding author name(s): Professor Robert Jinkerson

Editorial Notes:

Transferred manuscripts This manuscript has been previously reviewed at another journal. This document only contains reviewer comments, rebuttal and decision letters for versions considered at Nature Genetics.

Reviewer Comments & Decisions:

Decision Letter, initial version:

24th Jun 2021

Dear Professor Jinkerson,

Your Article, "Systematic characterization of gene function in a photosynthetic organism" has now been seen by 3 referees. You will see from their comments below that while they find your work of interest, some important points are raised. We are interested in the possibility of publishing your study in Nature Genetics, but would like to consider your response to these concerns in the form of a revised manuscript before we make a final decision on publication.

As you will see from these comments, all referees have identified aspects of the analyses and the methodology that need to be improved or clarified. Please address all referees' points as thoroughly as possible. We therefore invite you to revise your manuscript taking into account all reviewer and editor comments. Please highlight all changes in the manuscript text file. At this stage we will need you to upload a copy of the manuscript in MS Word .docx or similar editable format.

*2) If you have not done so already please begin to revise your manuscript so that it conforms to our Article format instructions, available [here](http://www.nature.com/ng/authors/article_types/index.html). Refer also to any guidelines provided in this letter.

[REDACTED]

We hope to receive your revised manuscript within eight to twelve weeks. If you cannot send it within this time, please let us know.

Sincerely,

Wei

Wei Li, PhD
Senior Editor
Nature Genetics
One New York Plaza, 47th Fl.
New York, NY 10004, USA
www.nature.com/ng

Reviewers' Comments:

Reviewer #1:

Remarks to the Author:

This paper reports a systematic phenotyping of a previously reported mutant collection of the model green alga *Chlamydomonas reinhardtii* (Ref. 12), which have been distributed to the community through the so-called CLiP library (<https://www.chlamylibrary.org/>). These mutants were barcoded, but this paper demonstrated its advantage for the first time in this study. The growth of each mutant in the pool is traceable by using these barcodes. Their growth under the various treatment was evaluated by the abundance of barcodes. The authors call the relative barcode abundances as the phenotype.

In this study, a new concept of systematic forward genetics was established. The *mars1* mutant was sensitive to a DNA damaging reagent, suggesting cpUPR could be activated upon DNA damage. On one hand, it demonstrated its applicability to relate a mutation to unexpected phenotype(s), but on the other, such information alone would not "guide for understanding the functions of thousands of poorly characterized genes". This mutant library would be beneficial to the community without doubt, but it would be difficult to evaluate such a paper.

There are at least a few points that should be addressed before publication.

One of the advantages of this system is that phenotypes are treated as quantitative measures and the authors claim that it is "the largest genotype-by-phenotype dataset", but did the authors confirm that those mutants only carry single mutations? This problem should be cleared. If the mutants possibly carry multiple mutations, deep-sequencing of each mutant would be necessary and the genotypes may be treated as more like QTL.

It is also not clear how much percentage of the genes were actually barcoded in their 3'UTR or introns, not in the CDS. In fact, many mutants seem to have a barcode in those non-coding regions (Table S4). When claiming the number of mutants, it is not fair to include those mutants. There are also only 684 high confident mutants identified in this study, which covers less than 5% of all genes in *C. reinhardtii*. This should be also clarified.

Reviewer #2:

Remarks to the Author:

A. Summary of the key results

The authors performed a genome-scale phenotyping in which they measured the growth of 58000 tagged *Chlamydomonas* mutants in 121 environmental and chemical stress conditions. They analyzed the growth defect of the mutants based on the abundance of the respective bar-code tagging in non-stress and stress condition. They validated their methodology by showing that well-characterized mutants show expected genotype - phenotype specificity. They associated 684 genes to a phenotype, including hundreds with no functional annotation and proposed 89 names for genes of unknown function. They provide examples of their new findings related to poorly characterized genes in DNA repair, photosynthesis, CO₂ concentrating mechanism, cilia function and cytoskeleton integrity.

B. Originality and significance

To my knowledge, this functional genome approach has not yet been realized in any photosynthetic eukaryote and is really impressive by the scale of the work, which includes the analysis of 58000 mutants covering 78% of all the genes of *Chlamydomonas*.

C. Data and methodology: validity of approach, quality of data, quality of presentation

The validity of the approach has been verified by replicates of the treatments, where the same mutants/genes are found.

Concerning the treatments, could the authors explain why they focus on growth inhibitors (LATCA) of *Arabidopsis*? Is it really something that will represent valuable data for the *Chlamydomonas* community?

Some precisions are missing concerning how the pools of the mutant library were realized. Is a specific cell number of each mutant line was used for pooled growth? What is an 'homogenous' culture? How many cells/mL? Why 8 divisions before the analysis? Is the stationary phase reached at that moment? Could the authors provide growth curves? It could be interesting to compare the severity of the different treatments.

Concerning the genotype-phenotype relationship, many mutants contain a cassette with only one side mapped, with low level of confidence (73%, 58%) or in UTR where the phenotype is expected to be very mild. Could the authors comment on this point? Is it possible to deal with this issue?

How many replicates per treatment?

Concerning the new gene assignments in photosynthesis, could the authors provide an explanation on the fact that Cre06.g282800 is a putative thioredoxin (DUF1995)?

How could the authors deal with possible wrong gene annotations of the genome?

Figure 2: c and b should be inverted.

D.

No comments

E. Conclusions: robustness, validity, reliability

Conclusions are sound.

The last paragraph (352-355) is maybe too vague. Considering the magnitude of the work completed (number of treatments), could the authors comment on the number of the new genes assigned to the phenotypes?

F. Suggested improvements, experiments

No additional experiments are required.

G. References are appropriate.

H. The clarity of the manuscript is excellent, the demonstration that this high throughput screening for the assignment of gene function at the genome scale is made.

Reviewer #3:

Remarks to the Author:

This manuscript provides a comprehensive analysis of gene function in *Chlamydomonas reinhardtii*, a unicellular alga that has emerged as a powerful model system for photosynthetic organisms. The authors have used a barcoded mutant library of this alga to establish a genotype to phenotype dataset from over 58'000 mutants under a wide range of different environmental and chemical stress conditions. An important outcome of this work is that it has allowed the authors to place both known and unknown genes into specific pathways such as DNA repair, photosynthesis, carbon concentrating mechanism and flagellar biogenesis.

The authors show that this new resource can be used for the identification of novel genes involved in these important processes not only in *Chlamydomonas* but also in land plants. It will thus be highly useful for research in plant genetics and more generally in cell biology.

The quality of this work is high and the conclusions of the manuscript are well supported by the data presented. The manuscript is clearly written.

My only concern is the large number of Supplementary Tables some of which are not easy to understand. The main message of some of these Tables could probably be summarized in a few sentences without the need of showing all the details.

Minor comments

I. 305: Fig 4g should be Fig. 4e

Author Rebuttal to Initial comments

Reviewer #1:

Remarks to the Author:

This paper reports a systematic phenotyping of a previously reported mutant collection of the model green alga *Chlamydomonas reinhardtii* (Ref. 12), which have been distributed to the community through the so-called CLiP library (<https://www.chlamylibrary.org/>). These mutants were barcoded, but this paper demonstrated its advantage for the first time in this study. The growth of each mutant in the pool is traceable by using these barcodes. Their growth under the various treatment was evaluated by the abundance of barcodes. The authors call the relative barcode abundances as the phenotype.

In this study, a new concept of systematic forward genetics was established. The *mars1* mutant was sensitive to a DNA damaging reagent, suggesting cpUPR could be activated upon DNA

damage. On one hand, it demonstrated its applicability to relate a mutation to unexpected phenotype(s), but on the other, such information alone would not “guide for understanding the functions of thousands of poorly characterized genes”. This mutant library would be beneficial to the community without doubt, but it would be difficult to evaluate such a paper.

There are at least a few points that should be addressed before publication.

We are pleased that the submitted manuscript was well-received and that the impact of the presented data on the broader scientific community is recognized. We appreciate the constructive input from Reviewer 1. We address the Reviewer’s points below.

One of the advantages of this system is that phenotypes are treated as quantitative measures and the authors claim that it is “the largest genotype-by-phenotype dataset”, but did the authors confirm that those mutants only carry single mutations? This problem should be cleared. If the mutants possibly carry multiple mutations, deep-sequencing of each mutant would be necessary and the genotypes may be treated as more like QTL.

We thank Reviewer 1 for these suggestions. Previous work from our group characterized the *Chlamydomonas* mutant library (Li *et al.* 2019, Ref. 12) and determined that the average number of mapped integration events per mutant to be 1.2. We have added a reference to this in the main text of the revision to emphasize this fact:

Line#123 of the main text: “When multiple independent mutant alleles for the same gene show the same phenotype, the confidence in a gene lesion-phenotype relationship increases because it is less likely that the phenotype is due to a mutation elsewhere in the genome (on average there are 1.2 cassette integration events per mutant, and the mutants can also carry other mutations such as point mutations), or that there was an error in mapping of the mutation¹².”

We agree with Reviewer #1 that multiple insertions or any other random unknown mutations, such as point mutations, can make it difficult to link a phenotype from a single mutant to a specific gene. For this reason, we differentiate between mutant-level and gene-level phenotypes. To connect a phenotype to a specific gene, we developed a statistical framework that leverages multiple independent mutant alleles in the same gene. We require a minimum of 3 independent mutant alleles for a gene to be included in the analysis. This allows us to be more confident in assigning gene-level phenotypes as multiple mutant alleles will have to exhibit a similar phenotype, negating any effects from other mutations that may be found in individual mutants. For genes with fewer than 3 multiple independent mutations, the mutant-level phenotype data we provide is robust, but the link to a specific gene would need to be validated.

This could be accomplished by traditional methods such as complementation or back-crossing. We clarified this in the main text:

Line#107 of the main text: “While a lone mutant showing a phenotype is not sufficient evidence to conclusively establish a gene-phenotype relationship, we anticipate that these data will be useful to the research community in at least three ways: first, they can help prioritize the characterization of candidate genes identified by other means, such as transcriptomics or protein-protein interactions. Second, they facilitate the generation of hypotheses about the functions of poorly characterized genes. Third, they enable prioritization of available mutant alleles for further studies, including to establish a gene-phenotype relationship by complementation and/or back-crossing.”

Line #81: We also changed in the main text: “Taken together, this effort represents, to the best of our knowledge, the largest mutant-by-phenotype dataset to date for any photosynthetic organism, with 16.8 million data points (Table S4).”

It is also not clear how much percentage of the genes were actually barcoded in their 3'UTR or introns, not in the CDS. In fact, many mutants seem to have a barcode in those non-coding regions (Table S4). When claiming the number of mutants, it is not fair to include those mutants. There are also only 684 high confident mutants identified in this study, which covers less than 5% of all genes in *C. reinhardtii*. This should be also clarified.

We thank Reviewer 1 for these suggestions.

The insertion cassette used to generate the mutants contains two transcriptional terminators, thus we reasoned that insertions in 5' UTRs, introns, and exons will lead to transcriptional disruption and loss of function mutants. Based on our previous work (see Fig. 3d of Li et al. 2019, Ref. 12), mutants with cassette integrations in the 3'UTR are less likely to produce phenotypes, so were excluded from our statistical framework. These mutants with insertions in 3'UTRs represent ~25% of the mapped integrations in the library (Li et al. 2019, Ref. 12). We clarified this in the main text, material and methods, and adjusted Fig. 2f pie chart:

Line#105 of the main text: “10,380 genes (59% of all *Chlamydomonas* genes) are represented by one or more 5'UTR, CDS, or intron insertion mutant that showed a phenotype (decreased abundance below our detection limit) in at least one screen (Fig. 2f).”

Line#690 of material and methods: “Only alleles that were mapped with confidence level 4 or less (corresponding to a likelihood of correct mapping of 58% or higher)¹², had an insertion in CDS/intron/5'UTR feature, and had greater than 50 reads in the control condition were included

in the analysis. The frequency of cassette insertion location based on gene feature was: intron 25%, 3'UTR 23%, CDS 19%, not mapped 14%, intergenic 6%, 5'UTR 5%, and multiple or others 8%. The insertion cassette used to generate the mutants contains two transcriptional terminators, one in each direction, thus we reasoned that insertions in 5'UTRs, introns, and exons will lead to transcriptional disruption and loss of function mutants. Mutants with cassette integrations in the 3'UTR were not expected to result in transcriptional disruption, so these mutants were excluded from our statistical framework."

Line#420 of the main text Figure 2 panel f: The pie chart was re-plotted to reflect the updated analysis which excludes 3'UTR.

Reviewer #2:

Remarks to the Author:

A. Summary of the key results

The authors performed a genome-scale phenotyping in which they measured the growth of 58000 tagged *Chlamydomonas* mutants in 121 environmental and chemical stress conditions. They analyzed the growth defect of the mutants based on the abundance of the respective barcode tagging in non-stress and stress condition. They validated their methodology by showing that well-characterized mutants show expected genotype - phenotype specificity. They associated 684 genes to a phenotype, including hundreds with no functional annotation and proposed 89 names for genes of unknown function. They provide examples of their new findings related to poorly characterized genes in DNA repair, photosynthesis, CO₂ concentrating mechanism, cilia function and cytoskeleton integrity.

B. Originality and significance

To my knowledge, this functional genome approach has not yet been realized in any photosynthetic eukaryote and is really impressive by the scale of the work, which includes the analysis of 58000 mutants covering 78% of all the genes of *Chlamydomonas*.

C. Data and methodology: validity of approach, quality of data, quality of presentation

The validity of the approach has been verified by replicates of the treatments, where the same mutants/genes are found.

We are pleased that the submitted manuscript was well-received. We appreciate the constructive input from Reviewer 2. We address the Reviewer's points below.

Concerning the treatments, could the authors explain why they focus on growth inhibitors (LATCA) of *Arabidopsis*? Is it really something that will represent valuable data for the *Chlamydomonas* community?

We thank Reviewer 2 for these questions. We reasoned that small molecule treatments would allow us to improve the clustering of genes based on their phenotypic profile across a broad variety of different conditions. We chose to screen the LATCA library for active compounds in Chlamydomonas because we believed that these compounds would be more likely to impact pathways both in Chlamydomonas and land plants, thus providing more general insights into gene functions in the green lineage. We have edited the text to provide additional context to address Reviewer 2's questions:

Line#78 of the main text: "To further expand the range of stressors in the dataset, we identified 1,222 small molecules from the Library of AcTive Compounds on Arabidopsis (LATCA)¹³ that negatively influence Chlamydomonas growth (Extended Data Fig. 1, Table S3, Extended Data File 1), and performed competitive growth experiments in the presence of 52 of the most potent compounds. We chose to screen the LATCA library for active compounds in Chlamydomonas because we believed that these compounds would be more likely to impact pathways both in Chlamydomonas and plants, thus providing more general insights into gene functions in the green lineage."

Some precisions are missing concerning how the pools of the mutant library were realized. Is a specific cell number of each mutant line was used for pooled growth? What is an 'homogenous' culture? How many cells/mL?

We thank Reviewer 2 for these suggestions. We agree with Reviewer 2 that some of these steps could be explained more precisely, therefore we added the following information in the manuscript:

Line#67 of the main text: "We pooled the entire Chlamydomonas mutant collection from plates into a liquid culture and used molecular barcodes to quantify the relative abundance of each mutant after competitive growth (Fig. 1a-f)."

We added a more technical description into the materials and methods to further clarify the details of the pooled growth experiments:

Line#595 of material and methods: "Cultures were inoculated with 2×10^4 cells ml^{-1} and most experiments were performed in 2 L vessels (4×10^7 cells total) with ~58k mutants (Table S1, S2), resulting in ~700 cells per mutant on average in the 2 L competitive growth experiments."

We also removed "homogenous" from the legend of Figure 1. We used the word "homogenous" to describe equal ratios of different mutants between different experiments, but we now feel that

this word introduces more confusion than clarity. In the material and methods sections we describe the underlying experimental procedure of breaking cell clumps after pooling and proper mixing of the mutant pools (line#582-588).

Why 8 divisions before the analysis? Is the stationary phase reached at that moment? Could the authors provide growth curves? It could be interesting to compare the severity of the different treatments.

We thank Reviewer 2 for these suggestions.

The basis of our pooled screening approach is a competitive growth experiment where mutants are grown under different treatment conditions. In order to assess phenotypes of either slower growth (or potentially none at all) or faster growth we needed to have enough cell divisions to allow a detection of this difference without reaching stationary phase where experiments are less reproducible. At the same time, we needed to start with enough cells so that the starting cell number for each mutant did not vary substantially between cultures due to sampling noise. We previously established that seven divisions is ideal (Li et al., 2019). The reviewer's comment helped us realize that there was a typo in Figure 1: panel b should read "after ~7 doublings"; this has now been corrected. We have also added the following sentence to the manuscript for further clarification:

Line#607 of the materials and methods: "We sought to avoid letting the cultures reach stationary phase, where experiments are less reproducible. At seven divisions the mutant pool was typically in the late exponential growth phase."

Due to the number of tested conditions we were unable to collect growth curve data for the entirety of the experiment. We visually tracked cell density and cell counts were obtained when experiments appeared to reach late exponential densities. The severity of the treatment can be estimated based on the number of mutants that are impacted by the growth condition (see Figure 2b). For example, micronutrient variations or LatB did not impact overall mutant growth much compared to some chemical treatments (e.g. DNA damage).

Concerning the genotype-phenotype relationship, many mutants contain a cassette with only one side mapped, with low level of confidence (73%, 58%) or in UTR where the phenotype is expected to be very mild. Could the authors comment on this point? Is it possible to deal with this issue?

We thank Reviewer 2 for these questions.

We recognize that many of the mutants have insertions with low-confidence mappings (and, as Reviewer 1 noted, some may carry additional mutations). We deal with these issues in two ways:

1) We have high confidence in the phenotyping data of the individual mutants, and provide these phenotypes as a starting point for the research community's characterization of putative gene-phenotype relationships using the available mutants. We added a note to clarify this in the main text (see also response to Reviewer 1):

Line#107 of the main text: "While a lone mutant showing a phenotype is not sufficient evidence to conclusively establish a gene-phenotype relationship, we anticipate that these data will be useful to the research community in at least three ways: first, they can help prioritize the characterization of candidate genes identified by other means, such as transcriptomics or protein-protein interactions. Second, they facilitate the generation of hypotheses about the functions of poorly characterized genes. Third, they enable prioritization of available mutant alleles for further studies, including to establish a gene-phenotype relationship by complementation and/or back-crossing."

2) For genes represented by three or more independent mutants, we used our statistical analysis pipeline to connect genes to phenotypes. The details of our statistical analysis pipeline are outlined in the Material and Methods section of this manuscript (line#677-699) and we referenced our previous work by Li et al. (2019) that includes a mutant feature analysis. We note that we removed 3'UTR mutants (as outlined above in our response to Reviewer 1) and mutants with a mapping confidence below 58% from the statistical analysis pipeline, as indeed, these tend to give mild phenotypes.

How many replicates per treatment?

We thank Reviewer 2 for this question. We had 146 different treatments, of which 49 treatments had two or more replicates, with some treatments having 4 or more replicates such as (number of replicates): TAP-Light (14), TP-Air (8), TP-CO₂ (8), TAP-Dark (6), Cisplatin (4), MMC (4), high temperature (4), and spectinomycin (4). A summarized list of all treatments can be found in Table S2. To clarify this point in the manuscript we changed the following sentence:

Line#598 of the materials and methods: "Cultures were grown under a broad variety of conditions (Table S2) of which 49 had two or more replicates."

Concerning the new gene assignments in photosynthesis, could the authors provide an explanation on the fact that Cre06.g282800 is a putative thioredoxin (DUF1995)?

We thank Reviewer 2 for pointing out the missannotation of Cre06.g281800. It is annotated as “domain of unknown function (DUF1995)” according to the Chlamydomonas genome annotation release 5.6. We have now corrected the mistake.

Line#207 of the main text now reads: “Several highly conserved but poorly characterized genes are also found in this cluster, including the putative Rubisco methyltransferase Cre12.g52450036, the putative thioredoxin Cre01.g037800, the predicted protein with a domain of unknown function (DUF1995) Cre06.g281800 (which we named LIGHT SENSITIVE AND/OR ACETATE-REQUIRING 4, LSAR4), and Cre13.g572100 (which we named LIGHT GROWTH SENSITIVE 4, LGS4);”

How could the authors deal with possible wrong gene annotations of the genome?

We thank Reviewer 2 for this question. Our analysis pipeline is independent of current gene annotations as we link phenotype to specific gene models. We did use the current gene annotations, Chlamydomonas 5.6, to validate our phenotypic data. For example we found that in many screens, mutants that exhibited phenotypes were enriched for disruptions in genes with expected function based on GO-Term annotations. Gene annotations are largely derived from homology analysis and incorrect gene annotation can stymie research efforts. Our phenotypic data can hopefully help validate gene annotations and identify those that may need to be reannotated.

Line#721 of the materials and methods text: “All analysis was performed using JGI Phytozome release v5.0 of the Chlamydomonas assembly and v5.6 of the Chlamydomonas annotation⁷⁶. Gene identifiers (CreXX.gXXXXXX) can be used to link data found in the supplemental tables to gene annotation updates.”

Additionally, the genomic location of all barcode insertions have been identified (more information can be found in Li et al. 2019) so in the future if gene models are updated, the analysis can be re-run to take these updates into account. High confidence phenotype data is also available on the Chlamydomonas Mutant Library website (<https://www.chlamylibrary.org/>) for ease of access to the community.

Figure 2: c and b should be inverted.

We thank Reviewer 2 for catching this. We had inverted c and b in the Figure 2 figure legend and this error has now been corrected.

D.

No comments

E. Conclusions: robustness, validity, reliability

Conclusions are sound.

We thank Reviewer 2 for this kind comment.

The last paragraph (352-355) is maybe too vague. Considering the magnitude of the work completed (number of treatments), could the authors comment on the number of the new genes assigned to the phenotypes?

We thank Reviewer 2 for this suggestion. We moved the following text from the results to the last paragraph of the discussion.

Line#354 of the discussion: “Altogether, 58% of the high-confidence gene-phenotype interactions involve a Chlamydomonas gene with a predicted Arabidopsis homologue (Table S8); for approximately 79% of the corresponding Arabidopsis homologues our data predict a novel gene-phenotype relationship.”

F. Suggested improvements, experiments

No additional experiments are required.

G. References are appropriate.

H. The clarity of the manuscript is excellent, the demonstration that this high throughput screening for the assignment of gene function at the genome scale is made.

We thank Reviewer 2 for these kind comments.

Reviewer #3:

Remarks to the Author:

This manuscript provides a comprehensive analysis of gene function in *Chlamydomonas reinhardtii*, a unicellular alga that has emerged as a powerful model system for photosynthetic organisms. The authors have used a barcoded mutant library of this alga to establish a genotype to phenotype dataset from over 58'000 mutants under a wide range of different environmental and chemical stress conditions. An important outcome of this work is that it has

allowed the authors to place both known and unknown genes into specific pathways such as DNA repair, photosynthesis, carbon concentrating mechanism and flagellar biogenesis. The authors show that this new resource can be used for the identification of novel genes involved in these important processes not only in *Chlamydomonas* but also in land plants. It will thus be highly useful for research in plant genetics and more generally in cell biology. The quality of this work is high and the conclusions of the manuscript are well supported by the data presented. The manuscript is clearly written.

We are pleased that the submitted manuscript was well-received and that the impact of the presented data on the entire plant community is recognized. We appreciate the constructive input from Reviewer 3. We address the Reviewer's points below.

My only concern is the large number of Supplementary Tables some of which are not easy to understand. The main message of some of these Tables could probably be summarized in a few sentences without the need of showing all the details.

We thank Reviewer 3 for this suggestion. We think that the supplementary tables are necessary to enable further analysis. We recognize that some of these tables are not easy to understand and not ideal for quick look-ups of e.g. the phenotype of a particular mutant under a specific condition. Hence, we have made the most useful data accessible to the community via the mutant and gene pages of the *Chlamydomonas* Resource Center website.

In addition, we reviewed the titles and legends of all supplementary tables and added the following changes to the supplementary tables to address Reviewer 3's comment. Some of these descriptions were previously included in the SI guide or technical legends of the tables.

- **Table S1. Source material used for screens.** The screening was done in six different rounds (R1-R6). These rounds differed in parameters such as how many mutants were used (the full original library or the library after rearray to eliminate mutants with unmapped insertions), or whether the mutants were pooled from 384 or 1536-colony array plates.
- **Table S2. List of all treatments.** The mutant pools were subjected to various different treatments. Specific treatment conditions are listed in this table while general procedures are described in the material and method section.
- **Table S3. LATCA screen and dose titration validation.** We identified LATCA compounds that inhibit *Chlamydomonas* growth in TAP and TP media and determined the final mutant screening concentration for selected compounds. The initial screen

included 3,650 LATCA chemicals to determine their effect on growth of cMJ030. We further validated 1,140 of these chemicals before selecting compounds for mutant screens.

- **Table S4. Mutant phenotypes across all screens.** Table of phenotype data collected for 94,461 mutants in 223 screens.
- **Table S5. FDRs for GO term enrichment.** This table shows GO term enrichment in mutants that show a phenotype in screens.
- **Table S6. FDRs for all genes in all screens.** Table of FDRs from our statistical framework to identify high-confidence gene-phenotype relationships of 15,582 genes in 223 screens.
- **Table S7. High-confidence gene-phenotype relationships.** Table of the gene-phenotype relationships with an FDR <0.3. The “Gene phenotype” is the median phenotype (log₂ ratio of the treatment versus the control) for all alleles of that gene that were included in the FDR calculation. A phenotype value of -10 was used when the median treatment had zero reads. Gene names with * were named in this study.
- For Tables S8, S9, S10, S11, S13 and S14 we moved descriptions from the table to a more prominent position in the table legend. Table S12’s description can be found in the SI guide only. Table S15 does not require additional information.

Minor comments

I. 305: Fig 4g should be Fig. 4e

We thank Reviewer 3 for catching this. We changed Fig 4g to Fig 4f in the main text.

Decision Letter, first revision:

Our ref: NG-A57663R

7th Oct 2021

Dear Dr. Jinkerson,

Thank you for submitting your revised manuscript "Systematic characterization of gene function in a photosynthetic organism" (NG-A57663R). It has now been seen by the original referees and their comments are below. The reviewers find that the paper has improved in revision, and therefore we'll be happy in principle to publish it in Nature Genetics, pending minor revisions to comply with our editorial and formatting guidelines.

Sincerely,

Wei

Wei Li, PhD
Senior Editor
Nature Genetics
New York, NY 10004, USA
www.nature.com/ng

Reviewer #1 (Remarks to the Author):

The authors have responded to all my comments in a satisfactory manner. As the authors responded, It is important to note that careful linking of a phenotype to a genotype cannot be overemphasized especially in this class of large scale study. I understand that there is a low possibility of multiple tag insertion or unidentified point mutations. In this regard, the presented statistic framework sounds reasonable. This study was overall carefully done and the manuscript reads very well. As I said, the library is extremely useful and the effort will be highly appreciated by the community for sure.

Reviewer #2 (Remarks to the Author):

The authors have clarified the points raised.

Reviewer #3 (Remarks to the Author):

I recommend acceptance of the revised version.

Author Rebuttal, first revision:

We are pleased that the submitted manuscript was well-received and that the impact of the presented data on the broader scientific community is recognized. We appreciate the constructive input from all three reviewers throughout the review process.

Reviewer #1 (Remarks to the Author):

The authors have responded to all my comments in a satisfactory manner. As the authors responded, It is important to note that careful linking of a phenotype to a genotype cannot be overemphasized especially in this class of large scale study. I understand that there is a low possibility of multiple tag insertion or unidentified point mutations. In this regard, the presented statistic framework sounds reasonable. This study was overall carefully done and the manuscript reads very well. As I said, the library is extremely useful and the effort will be highly appreciated by the community for sure.

Reviewer #2 (Remarks to the Author):

The authors have clarified the points raised.

Reviewer #3 (Remarks to the Author):

I recommend acceptance of the revised version.

Final Decision Letter:

In reply please quote: NG-A57663R1 Jinkerson

15th Mar 2022

Dear Dr. Jinkerson,

I am delighted to say that your manuscript "Systematic characterization of gene function in the photosynthetic alga *Chlamydomonas reinhardtii*" has been accepted for publication in an upcoming issue of Nature Genetics.

Your paper will be published online after we receive your corrections and will appear in print in the next available issue. You can find out your date of online publication by contacting the Nature Press Office (press@nature.com) after sending your e-proof corrections. Now is the time to inform your Public Relations or Press Office about your paper, as they might be interested in promoting its publication. This will allow them time to prepare an accurate and satisfactory press release. Include your manuscript tracking number (NG-A57663R1) and the name of the journal, which they will need when they contact our Press Office.

Acceptance is conditional on the data in the manuscript not being published elsewhere, or announced in the print or electronic media, until the embargo/publication date. These restrictions are not intended to deter you from presenting your data at academic meetings and conferences, but any

enquiries from the media about papers not yet scheduled for publication should be referred to us.

Please note that *Nature Genetics* is a Transformative Journal (TJ). Authors may publish their research with us through the traditional subscription access route or make their paper immediately open access through payment of an article-processing charge (APC). Authors will not be required to make a final decision about access to their article until it has been accepted. [Find out more about Transformative Journals](https://www.springernature.com/gp/open-research/transformative-journals)

Authors may need to take specific actions to achieve [compliance with funder and institutional open access mandates](https://www.springernature.com/gp/open-research/funding/policy-compliance-faqs). If your research is supported by a funder that requires immediate open access (e.g. according to [Plan S principles](https://www.springernature.com/gp/open-research/plan-s-compliance)) then you should select the gold OA route, and we will direct you to the compliant route where possible. For authors selecting the subscription publication route, the journal's standard licensing terms will need to be accepted, including [self-archiving-and-license-to-publish](https://www.nature.com/nature-portfolio/editorial-policies/self-archiving-and-license-to-publish). Those licensing terms will supersede any other terms that the author or any third party may assert apply to any version of the manuscript.

Please note that Nature Research offers an immediate open access option only for papers that were first submitted after 1 January, 2021.

If you have not already done so, we invite you to upload the step-by-step protocols used in this

manuscript to the Protocols Exchange, part of our on-line web resource, natureprotocols.com. If you complete the upload by the time you receive your manuscript proofs, we can insert links in your article that lead directly to the protocol details. Your protocol will be made freely available upon publication of your paper. By participating in natureprotocols.com, you are enabling researchers to more readily reproduce or adapt the methodology you use. Natureprotocols.com is fully searchable, providing your protocols and paper with increased utility and visibility. Please submit your protocol to <https://protocolexchange.researchsquare.com/>. After entering your nature.com username and password you will need to enter your manuscript number (NG-A57663R1). Further information can be found at <https://www.nature.com/nature-portfolio/editorial-policies/reporting-standards#protocols>

Sincerely,

Wei Li, PhD
Senior Editor
Nature Genetics
New York, NY 10004, USA
www.nature.com/ng